# Role of hepatocyte RIPK1 in maintaining liver homeostasis during metabolic challenges

Weigao Zhang[1], Hu Liu[1,2], Danyang Zhang[1], Yuguo Yi[3], Liang Tao[4], Yunfeng Zhu[1], Shuxian Huang[1], Xunan Zhao[1], Qianchao Shao[1], Peiqi Li[1], Yiwen Weng[5], Wei Lu[6], Jianfa Zhang[1], Haibing Zhang[7]*, Yuxin Chen[2]*, Dan Weng[1]*

[1]School of Environmental and Biological Engineering, Key Laboratory of Metabolic Engineering and Biosynthesis Technology, Ministry of Industry and Information Technology, Nanjing University of Science and Technology, Nanjing, China; [2]Department of Laboratory Medicine, Nanjing Drum Tower Hospital, Affiliated Hospital of Medical School, Nanjing University, Nanjing, China; [3]School of Medicine, Shenzhen Campus of Sun Yat-sen University, Sun Yat-sen University, Shenzhen, China; [4]The First Affiliated Hospital, Basic Medical Sciences, University of South China, Hengyang, China; [5]Internal Medicine Department, Chengdu Jinniu District People's Hospital, Chengdu, China; [6]Affiliated Hospital of Nanjing University of Chinese Medicine, Nanjing, China; [7]CAS Key Laboratory of Nutrition, Metabolism and Food Safety, Shanghai Institute of Nutrition and Health, Shanghai Institutes for Biological Sciences, University of Chinese Academy of Sciences, Shanghai, China

*For correspondence:
hbzhang@sibs.ac.cn (HZ);
yuxin.chen@nju.edu.cn (YC);
danweng@njust.edu.cn (DW)

## eLife Assessment

This **important** study provides insights into the physiological role of RIPK1 in liver physiology, particularly during short-term fasting. The discovery that RIPK1 deficiency sensitizes the liver to acute injury and hepatocyte apoptosis is based on **convincing** evidence, highlighting the importance of RIPK1 in maintaining liver homeostasis under metabolic stress. The work will be of relevance to anyone studying liver pathologies.

**Abstract** As a central hub for metabolism, the liver exhibits strong adaptability to maintain homeostasis in response to food fluctuations throughout evolution. However, the mechanisms governing this resilience remain incompletely understood. In this study, we identified Receptor interacting protein kinase 1 (RIPK1) in hepatocytes as a critical regulator in preserving hepatic homeostasis during metabolic challenges, such as short-term fasting or high-fat dieting. Our results demonstrated that hepatocyte-specific deficiency of RIPK1 sensitized the liver to short-term fasting-induced liver injury and hepatocyte apoptosis in both male and female mice. Despite being a common physiological stressor that typically does not induce liver inflammation, short-term fasting triggered hepatic inflammation and compensatory proliferation in hepatocyte-specific RIPK1-deficient (*Ripk1*-hepKO) mice. Transcriptomic analysis revealed that short-term fasting oriented the hepatic microenvironment into an inflammatory state in *Ripk1*-hepKO mice, with up-regulated expression of inflammation and immune cell recruitment-associated genes. Single-cell RNA sequencing further confirmed the altered cellular composition in the liver of *Ripk1*-hepKO mice during fasting, highlighting the increased recruitment of macrophages to the liver. Mechanically, our results indicated that ER stress was involved in fasting-induced liver injury in *Ripk1*-hepKO mice. Overall, our findings revealed the role of RIPK1 in maintaining liver homeostasis during metabolic

fluctuations and shed light on the intricate interplay between cell death, inflammation, and metabolism.

## Introduction

During the long evolutionary history, mammals have frequently encountered various metabolic stresses arising from fluctuations in food availability. The liver, functioning as the central hub for metabolism, exhibits remarkable adaptability in maintaining both hepatic and systemic homeostasis when faced with these metabolic challenges (*Bernardo-Seisdedos et al., 2021*). Acute fasting, as one of the main causes of metabolic stress, is known to trigger hepatic glycogen depletion, increased production of hepatic glucose and ketone bodies, adipose tissue lipolysis, and the influx and accumulation of lipids in the liver (*Inokuchi-Shimizu et al., 2014*; *Petersen et al., 2017*; *Zoncu et al., 2011*). Under normal circumstances, the liver possesses the capability to shield itself from hepatic lipotoxicity caused by the elevated lipid influx and accumulation during fasting (*Cotter et al., 2014*; *Fromenty and Roden, 2023*). Nevertheless, the underlying mechanisms governing the liver's adaptive capacity to counteract lipotoxicity induced by metabolic stress are not fully understood.

The serine/threonine kinase RIPK1 is a crucial mediator of cell death and inflammation (*Mifflin et al., 2020*; *Zhao et al., 2023*). It manifests two distinct, even opposing functions: its scaffold function regulates cell survival and activation of NF-κB pathway, while its kinase activity promotes cell death including apoptosis and necroptosis (*Clucas and Meier, 2023*; *Ofengeim and Yuan, 2013*). Both functions play crucial regulatory roles across a spectrum of physiological and pathological scenarios. Notably, the kinase activity of RIPK1 has been extensively studied due to its association with the deleterious effects in pathological situations (*Majdi et al., 2020*; *Tao et al., 2021*; *Xu et al., 2021*; *Yin et al., 2022*). In contrast, the scaffold function of RIPK1 is less studied and current evidences suggest that it plays an essential role in maintaining tissue homeostasis within physiological context (*Mei et al., 2021*; *Najafov et al., 2021*). The scaffold function of RIPK1 has been studied using *Ripk1*$^{-/-}$ cells or RIPK1-deficient mice (*Kelliher et al., 1998*; *Mei et al., 2021*; *Najafov et al., 2021*; *Rickard et al., 2014*; *Silke et al., 2015*). The postnatality death of *Ripk1*$^{-/-}$ mice suggested that the scaffold function of RIPK1 plays an essential role in normal development, acting as a brake to prevent cell death and inflammation in different tissues (*Kelliher et al., 1998*; *Rickard et al., 2014*; *Silke et al., 2015*). Deletion of RIPK1 results in the loss of its scaffold function and unleashes the release of subsequent deleterious part with cell death and inflammation. Generation of conditional knockout mice with RIPK1 deletion in different tissues or cell types results in different phenotypes, suggesting that the role of RIPK1 scaffold in different tissues is dependent on the context in tissue or cells (*Dannappel et al., 2014*; *Filliol et al., 2016*; *Imanishi et al., 2023*; *Lu et al., 2023*). In contrast to the severe phenotype observed in mice with RIPK1 deletion in epithelial cells, mice lacking RIPK1 in hepatocytes (*Ripk1*-hepKO) exhibit a surprising state of health and normal under steady conditions (*Dannappel et al., 2014*; *Filliol et al., 2016*). Nonetheless, when confronted with pathogen-associated molecular patterns (PAMPs) or viral challenges, *Ripk1*-hepKO mice display heightened inflammation and cell death (*Farooq et al., 2019*; *Filliol et al., 2017*; *Filliol et al., 2016*). However, the precise role of hepatocellular RIPK1 in liver physiology remains unknown.

This study aimed to investigate the physiological functions of RIPK1 in the liver. Hepatocyte-specific RIPK1-deficient mice (*Ripk1*-hepKO) were generated and subjected to a 12-hr fasting period. Surprisingly, we observed that RIPK1 deficiency in hepatocytes sensitized the liver to acute liver injury and hepatocyte apoptosis. Remarkably, both male and female *Ripk1*-hepKO mice exhibited fasting-induced liver injury and apoptosis, emphasizing the robustness of this phenotype across genders. Furthermore, short-term fasting triggered hepatic inflammation in *Ripk1*-hepKO mice, as indicated by elevated expression of inflammatory markers and increased compensatory proliferation, potentially linked to hepatocellular carcinoma markers.

Transcriptomic profiling of liver tissues revealed that RIPK1 deficiency amplified the proinflammatory response during fasting, with up-regulated expression of inflammation-associated genes and enhanced recruitment of immune cells to the liver. Single-cell RNA sequencing further dissected the altered cellular composition in *Ripk1*-hepKO mice, highlighting an expansion of recruited macrophages (RMs) and NK&T cells. Sub-clustering of macrophages unveiled an increased presence of RMs in *Ripk1*-hepKO mice during fasting, suggesting their role in driving the fasting-induced inflammatory

state. In summary, our study sheds light on the critical role of RIPK1 in maintaining liver homeostasis during short-term fasting and provides insights into the intricate interplay between cell death, inflammation, and liver adaptation to metabolic challenges.

## Results

### RIPK1 deficiency in hepatocytes sensitizes the liver to short-term fasting-induced liver injury and hepatocyte apoptosis

To investigate the physiological functions of RIPK1 in liver homeostasis, hepatocyte-specific RIPK1-deficient mice (*Ripk1*-hepKO mice) were generated by crossing *Ripk1*<sup>flox/flox</sup> mice (*Ripk1*-Con mice) with *Albumin*-Cre transgenic mice. Deletion of RIPK1 in hepatocytes was confirmed by the analysis of RIPK1 expression in the liver tissue (*Figure 1—figure supplement 1A*). Consistent with previous reports (*Filliol et al., 2016*), *Ripk1*-hepKO mice were normal and healthy. Then both *Ripk1*-hepKO mice and their control littermates were subjected to short-term 12-hr fasting from 8:00 p.m. to 8:00 a.m. To our surprise, short-term fasting induced a significant increase in serum transaminase (alanine amino-transferase (ALT) and aspartate amino-transferase (AST)) levels in *Ripk1*-hepKO mice but not in their wild-type controls, suggesting that RIPK1 deficiency sensitizes the liver to short-term fasting, inducing acute liver injury (*Figure 1A, B*). Since RIPK1 is a key player regulating cell death including caspase-8-mediated apoptosis and RIPK3-MLKL-mediated necroptosis, we next examined whether the liver injury was due to cell apoptosis or necroptosis. As *Figure 1C, D* shown, the number of TUNEL-positive cells in the liver of *Ripk1*-hepKO mice was significantly higher compared to the control mice. Cleaved caspase-3 and phospho-MLKL are widely recognized as key proteins involved in the execution of apoptosis and necroptosis, respectively, and are commonly used as markers to detect these types of cell death. However, we were unable to detect the expression of cleaved caspase-3 and phospho-MLKL using the western blot, possibly due to their low expression levels (*Figure 1—figure supplement 1A*). Instead, we employed an immunofluorescence assay, which might be more sensitive for such detections. As indicated in *Figure 1E, F*, the number of cleaved caspase-3-positive cells was significantly higher in *Ripk1*-hepKO mice compared to controls. In contrast, we did not observe any change in the number of phospho-MLKL-positive cells in *Ripk1*-hepKO mice (*Figure 1G*). These results suggest that fasting-induced acute liver injury in *Ripk1*-hepKO mice might be due to apoptosis rather than necroptosis. To further investigate which cell types were undergoing apoptosis, we conducted the co-staining with TUNEL and Alb immunofluorescence, and the results demonstrated that most of TUNEL-positive cells co-localized with Alb-positive staining (*Figure 1H*), suggesting that short-term fasting-induced apoptosis mainly in hepatic parenchymal cells in *Ripk1*-hepKO mice.

It is known that fasting induces hypoglycemia and adipose tissue lipolysis, leading to the release of free fatty acids (FFA), which are transported to liver to facilitate the production of ketone bodies through fatty acid beta oxidation. We next investigated whether RIPK1 deficiency in hepatocytes affected the metabolic parameters during fasting. As *Figure 1I–O* shown, short-term fasting significantly decreased the blood glucose level, elevated the plasma levels of triglycerides (TG), total cholesterol (TC), FFA, β-hydroxybutyrate, and hepatic TG, and there was no obvious difference between wild-type control and *Ripk1*-hepKO mice, suggesting that RIPK1 deficiency did not affect the lipid metabolism in liver during fasting.

We next investigated whether sex difference affects fasting-induced acute liver injury in *Ripk1*-hepKO mice. Female *Ripk1*-Con mice and their *Ripk1*-hepKO littermates were subjected to short-term fasting. Consistent with male mice, we observed a significant increase in serum ALT/AST levels and hepatic TUNEL-positive cells in *Ripk1*-hepKO female mice (*Figure 1—figure supplement 1B–F*), suggesting that RIPK1 deficiency sensitizes the liver to short-term fasting-induced liver injury and hepatocyte apoptosis in both male and female mice.

Considering the complex and contrasting roles of RIPK1's scaffold function and kinase activity, to investigate whether the kinase activity of RIPK1 also contributes to maintaining hepatic homeostasis during fasting, RIPK1 kinase-dead (*Ripk1*-K45A) mice were utilized and subjected for short-term fasting. As shown in *Figure 1—figure supplement 2A, B*, we observed no increase in serum ALT/AST levels in *Ripk1*-K45A mice after 12 hr of fasting, suggesting that short-term fasting did not cause acute liver injury in these mice, unlike in *Ripk1*-hepKO mice. Additionally, short-term fasting resulted in elevated plasma levels of TG, TC, and hepatic TG, with no significant differences between wild-type

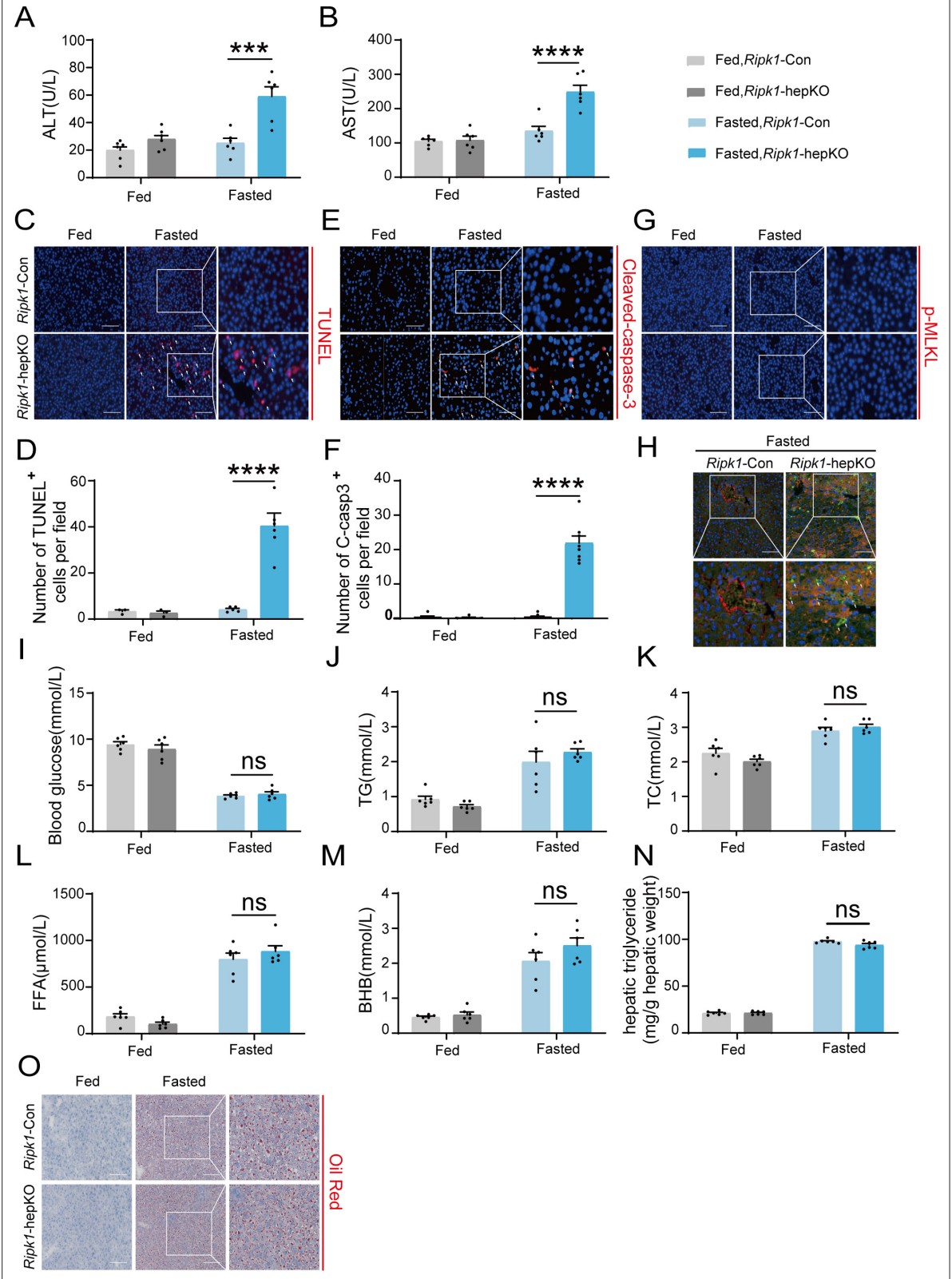

**Figure 1.** RIPK1 deficiency in hepatocytes sensitizes the liver to short-term fasting-induced liver injury and hepatocyte apoptosis. (**A**) Serum alanine amino-transferase (ALT) levels. (**B**) Serum aspartate amino-transferase (AST) levels. (**C, D**) Representative images and quantification of TUNEL staining. Scale bar, 100 μm. (**E, F**) Representative images and quantification of liver sections stained with anti-cleaved caspase-3 antibody (red) and DAPI (blue). Scale bar, 100 μm. (**G**) Representative images of liver sections stained with anti-phospho-MLKL antibody (red) and DAPI (blue). Scale bar,

*Figure 1 continued on next page*

*Figure 1 continued*

100 µm. (**H**) Fluorescence microscopy images of the liver stained with anti-ALB antibody (red), TUNEL (green), and DAPI (blue). Scale bar, 100 µm. (**I**) Blood glucose levels. (**J**) Serum triglycerides (TG) levels. (**K**) Serum total cholesterol (TC) levels. (**L**) Serum free fatty acids (FFA) levels. (**M**) Serum β-hydroxybutyrate (BHB) levels. (**N**) Hepatic TG levels (mg/g tissue). (**O**) Liver tissue was stained by Oil Red O. Scale bar, 100 µm. The data was analyzed via two- or one-way ANOVA. Data are expressed as mean ± SEM ($n$ = 6 per group). Asterisks denote statistical significance. ns, no significant, ***$p <$ 0.001, ****$p <$ 0.0001.

The online version of this article includes the following source data and figure supplement(s) for figure 1:

**Figure supplement 1.** RIPK1 deficiency in hepatocytes sensitizes the liver to short-term fasting-induced liver injury and hepatocyte apoptosis in female mice.

**Figure supplement 1—source data 1.** PDF file containing original western blots for *Figure 1—figure supplement 1A*, indicating the relevant bands and treatments.

**Figure supplement 1—source data 2.** Original files for western blot analysis displayed in *Figure 1—figure supplement 1A*.

**Figure supplement 2.** Analysis of *Ripk1*-WT and *Ripk1*-K45A mice before and after a 12-hr fasting period.

controls and *Ripk1*-K45A mice (*Figure 1—figure supplement 2C–E*). This suggests that the inactivation of RIPK1 kinase does not affect lipid metabolism in the liver during fasting and this was consistent with *Ripk1*-hepKO mice.

## Short-term fasting-induced hepatic inflammation and compensatory proliferation in *Ripk1*-hepKO mice

As a central hub for metabolism and many other physiological processes, liver coordinates a series of adaptations to maintain tissue homeostasis during food and energy fluctuations. Normally, short-term or temporary fasting will not disrupt liver homeostasis and cause tissue cell death and inflammation. However, consistent with liver injury and hepatocyte death, short-term fasting-induced hepatic inflammation in *Ripk1*-hepKO mice. As demonstrated in *Figure 2A*, the transcriptional expression of inflammatory markers, including *Ccl2*, *Tnfa*, *Ifng*, and *Il6*, was all significantly induced in *Ripk1*-hepKO mouse liver tissue, in contrast to wild-type littermates. We also examined the

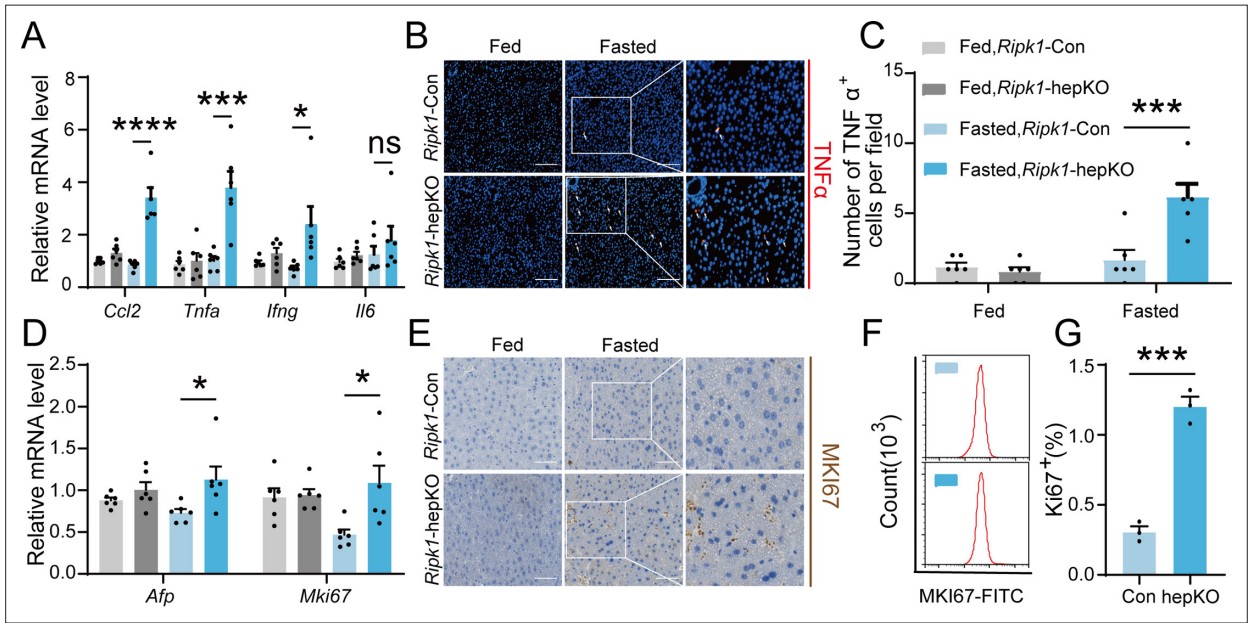

**Figure 2.** Short-term fasting-induced hepatic inflammation and compensatory proliferation in *Ripk1*-hepKO mice. (**A**) Hepatic mRNA expression of the inflammatory molecules. (**B, C**) Representative images and quantification of the liver stained with anti-TNFα antibody (red) and DAPI (blue). Scale bar, 100 µm. (**D**) Transcriptional expression of *Afp* and *Mki67* in liver tissue. (**E**) Representative images of the liver stained with anti-MKI67 antibody (brown) and hematoxylin (blue). Scale bar, 50 µm. (**F**) Representative flow cytometry plots showing MKI67-positive cells in the liver tissue ($n$ = 3 per group). (**G**) Relative increase of MKI67-positive cells in the liver tissue of *Ripk1*-hepKO mice after 12 hr of fasting compared to that of *Ripk1*-Con mice ($n$ = 3 per group). The data was analyzed via two- or one-way ANOVA. Data are expressed as mean ± SEM ($n$ = 6 per group). ns, no significant, *$p <$ 0.05, ***$p <$ 0.001, ****$p <$ 0.0001.

protein level of TNFα by histological staining (*Figure 2B, C*). As liver damage is known to activate compensatory proliferation, which is thought to promote hepatocarcinogenesis (*Inokuchi et al., 2010*; *Yang et al., 2013*), we also evaluated the expression of *Afp*, a marker gene of hepatocellular carcinoma, and *Mki67*, a proliferation-related antigen, and found a significant increase in both markers in *Ripk1*-hepKO mice after fasting (*Figure 2D*). To validate these findings at the protein level, we used immunohistochemistry of MKI67 to measure the rate of compensatory proliferation in livers and found a significant increase in the number of MKI67-positive cells in *Ripk1*-hepKO mice after 12 hr of fasting when liver injury was detected (*Figure 2E*). Similarly, flow cytometric analysis of liver tissue from mice after 12 hr of fasting also revealed a significant increase in the number of MKI67-positive cells in *Ripk1*-hepKO mice compared to control mice (*Figure 2F, G*). In contrast, no changes were observed in the transcriptional expression of inflammatory markers (*Ccl2*, *Tnfa*, *Ifng*, and *Il6*) or compensatory proliferation markers (*Afp* and *Mki67*) in the liver tissue of *Ripk1*-K45A mice during fasting (*Figure 1—figure supplement 2F, G*). Taken together, these results suggested that short-term fasting ignited inflammation and caused compensatory proliferation in *Ripk1*-hepKO mice.

## Transcriptomic profile of the mouse liver tissue upon short-term fasting

We next analyzed the liver transcriptome profiles to further characterize how short-term fasting oriented the hepatic microenvironment into an inflammatory state in *Ripk1*-hepKO mice. Consistent with previous reports, fasting significantly remodeled the transcriptional expression of various genes associated with metabolism in wild-type mice, among which 454 genes were up-regulated and 545 genes were down-regulated (*Figure 3—figure supplement 1A*). As expected, pathway enrichment (Kyoto Encyclopedia of Genes and Genomes, KEGG) analysis revealed that these differentially expressed genes (DEGs) were primarily enriched in metabolism-associated pathways, including fatty acid degradation, PPAR signaling pathway and MAPK signaling pathway (*Figure 3—figure supplement 1B*). Cluster analysis showed that fasting significantly altered metabolism-related genes in *Ripk1*-Con mice (*Figure 3—figure supplement 1C*). To be noteworthy, in wild-type mice, fasting normally suppressed the expression of immune- or inflammation-related genes, in agreement with the shifted energy to guarantee the need for basic survival other than immune protection during food deprivation. RIPK1 deficiency in hepatocytes did not greatly affect the gene expression profiles when food was available and there were 25 up-regulated and 54 down-regulated genes which were mostly associated with alcoholic liver disease, vascular smooth muscle contraction and cardiomyopathy in *Ripk1*-hepKO mice compared to wild-type control at fed state (*Figure 3—figure supplement 1D, E*). However, when subjected to metabolic fluctuation with short-term fasting, loss of RIPK1 in hepatocytes amplified the affect, and there were additional 85 up-regulated and 29 down-regulated genes identified in *Ripk1*-hepKO mice compared to the *Ripk1*-Con group (p < 0.05, |log2(fold change)| ≥ 1) (*Figure 3A*). Pathway enrichment (KEGG) analysis revealed that these DEGs were primarily enriched in inflammation-associated pathways, including TNF signaling pathway, IL-17 signaling pathway, toll-like receptor signaling pathway, NF-κB signaling pathway, etc. (*Figure 3B*). Cluster analysis of the DEGs revealed that hepatocyte-specific deletion of RIPK1 significantly induced the expression of many genes involved in the inflammatory process after fasting, including *Ccl2*, *Clec4e*, *Ikbke*, *Tnfaip3*, etc. (*Figure 3C*). Through further analysis of the transcriptome data, we observed a significant increase in the expression chemokine-related genes involved in the recruitment of immune cells, such as *Itgam*, *Ccr2*, and *Cx3cr1* (*Figure 3C*). To validate these findings at the protein level, we conducted immunofluorescence staining for F4/80, a marker for liver-resident immune cells, and CD11b, CCR2, and CX3CR1, markers for liver-recruited immune cells. Our results demonstrated a significant increase in the number of CD11b-, CCR2-, and CX3CR1-positive cells in *Ripk1*-hepKO mice after fasting, while F4/80 expression remained unchanged (*Figure 3D, E*). These results suggested that in contrast to the immunosuppression state of liver during fasting, RIPK1 deficiency sensitize the liver into a proinflammatory state, with induced expression of inflammation and immune cell recruitment-associated genes. Hepatic gene expression profile reflected the dynamics of RMs into liver in *Ripk1*-hepKO mice. RIPK1 might be act as an important protein to maintain the immune tolerant homeostasis in liver during metabolism changes.

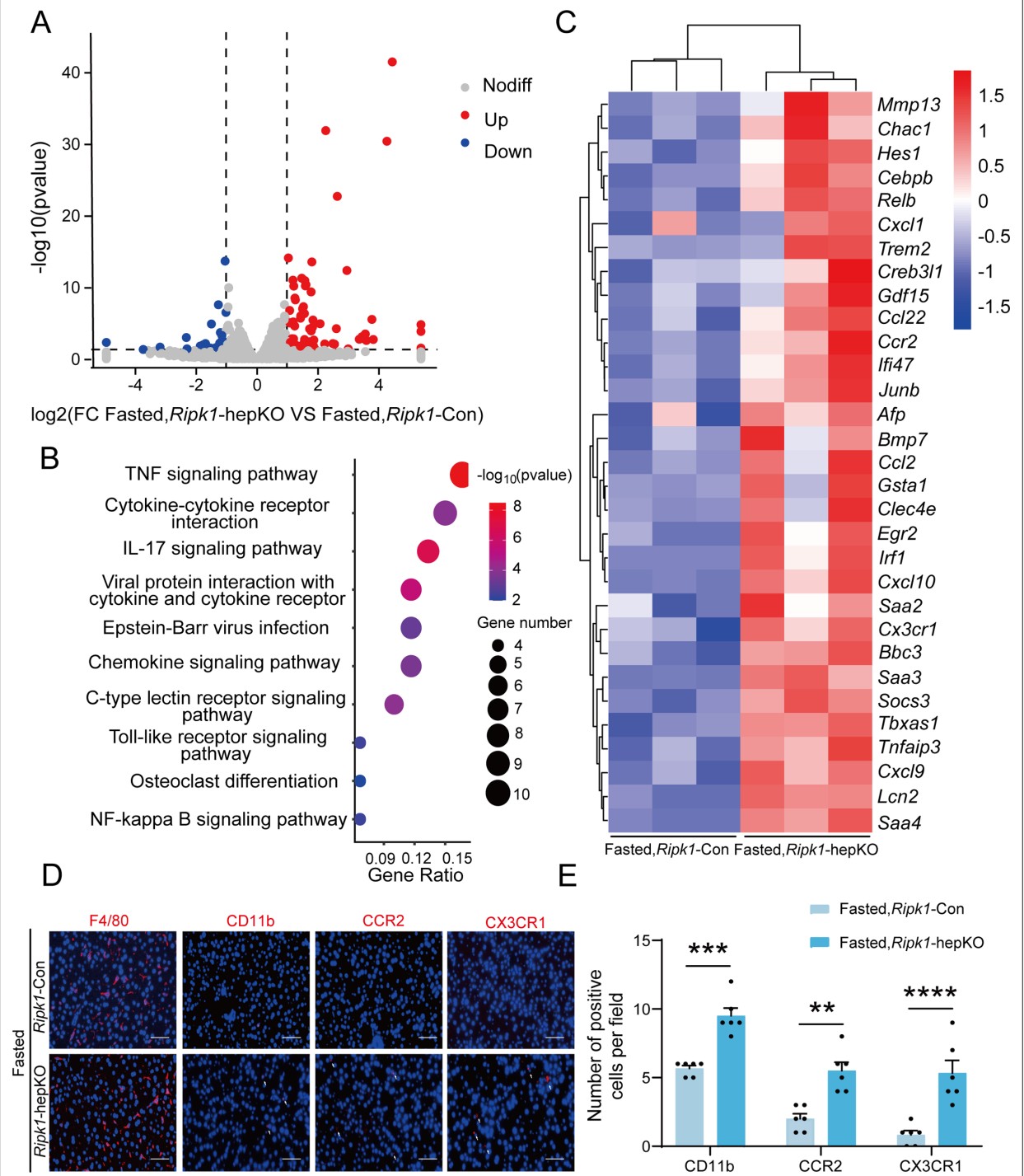

**Figure 3.** Transcriptome sequencing of the liver tissue from *Ripk1*-Con and *Ripk1*-hepKO mice. (**A**) The volcano plot of differentially expressed genes was illustrated. The blue spots represent the down-regulated genes in *Ripk1*-hepKO group compared with control (*Ripk1*-Con) group, and the red spots represent the up-regulated genes in *Ripk1*-hepKO group. (**B**) The altered signaling pathways were enriched by Kyoto Encyclopedia of Genes and Genomes (KEGG) analysis. (**C**) The genes which expression were significantly altered in *Ripk1*-hepKO group were depicted in the heatmap. (**D**, **E**) Representative fluorescence microscopy images and quantification of the liver stained with anti-F4/80 antibody (red), anti-CD11b antibody (red), anti-CCR2 antibody (red), and anti-CX3CR1 antibody (red), and DAPI (blue), respectively. Scale bar, 100 μm. The data was analyzed via two- or one-way ANOVA. Data are expressed as mean ± SEM (*n* = 6 per group). ns, no significant, **p < 0.01, ***p < 0.001, ****p < 0.0001.

The online version of this article includes the following figure supplement(s) for figure 3:

**Figure supplement 1.** Transcriptome sequencing of the liver tissue from *Ripk1*-Con and *Ripk1*-hepKO mice.

## Single-cell RNA sequencing analysis of the mouse liver tissue

Transcriptomic profiling of whole liver tissue revealed that hepatic inflammation induced by fasting in *Ripk1*-hepKO mice was likely arising from enhanced recruitment of immune cells to the liver. To confirm this hypothesis, we performed single-cell RNA sequencing of liver non-parenchymal cells isolated from *Ripk1*-hepKO or *Ripk1*-Con mice upon fasting. A total of 22,274 single-cell transcriptomes (10,374 *Ripk1*-Con; 11,900 *Ripk1*-hepKO) were obtained, and 30 major clusters were identified by T-distributed stochastic neighbor embedding (t-SNE) visualization, representing different cell types based on marker gene expression (*Figure 4—figure supplement 1A, B*). As the result shown, short-term fasting led to significant changes in the cellular composition of the liver in *Ripk1*-hepKO mice, characterized by an increase in the number of cells in clusters 12, 14, 19, 22, and 28 and a decrease in the number of cells in clusters 9, 11, 22, 26, and 29, compared to control mice (*Figure 4—figure supplement 1C, D*). In order to further explore the significance of these changes, we identified eight major cellular clusters in 12-hr fasted control liver, including B cells (*Cd19, Cd79a, Fcmr, Cd79b, Ebf1*), neutrophils (*S100a8, Csf3r, S100a9, Mmp9*), ECs (*Igfbp7, Ptprb, Clec4g, Kdr*), PCs (*Sdc1, CD138*), DCs (*Ly6d, Siglech, Rnase6, H2-Ab1*), NK&T cells (*Nkg7, Xcl1, Cd3d, Gzma*), macrophages (*Adgre1, Csf1r, Sdc3, Ifitm2*), and hepatocytes (*Alb, Saa1, Apoc1, Mup20*) (*Figure 4A*; *Sun et al., 2022*; *Zhao et al., 2020*). As the result shown, the numbers of macrophages and NK&T cells were increased in the liver of *Ripk1*-hepKO mice compared to that of *Ripk1*-Con mice upon short-term fasting, suggesting the dynamics of recruited immune cells into liver in *Ripk1*-hepKO mice (*Figure 4B*). We next focused on liver macrophages, which might contribute to the fasting-induced inflammatory state in *Ripk1*-hepKO mice.

As expected, pathway enrichment (KEGG) analysis based on macrophage differential genes revealed that these DEGs were primarily enriched in inflammation-associated pathways, including IL-17 signaling pathway, antigen processing and presentation and TNF signaling pathway, etc. (*Figure 4C*). We performed sub-clustering of macrophages to further dissect changes in the landscape of macrophages upon RIPK1 deficiency in hepatocytes. This cluster can be further divided into two groups of cells representing liver-resident macrophages (LrMs: KCs) and RMs based on their marker gene expression profile (*Figure 4D*). LrMs were characterized by high expression of *Adgre1* (encoding F4/80) and *Clec4f*, whereas RMs exhibited high expression of *Itgam* (encoding Cd11b, an important marker gene of bone marrow-derived macrophages), *Ccr2* (a chemokine receptor important for infiltration of circulating monocytes), and *Cx3cr1* (*Krenkel and Tacke, 2017*). We observed that while the number of LrMs remained almost unchanged in *Ripk1*-hepKO mice after short-term fasting compared to *Ripk1*-Con mice, the number of RMs significantly increased (*Figure 4E*). This was further confirmed by flow cytometry analysis, which revealed a significant increase in the proportion of F4/80+CD11b+ cells among all F4/80+ cells in *Ripk1*-hepKO mice compared to control mice (*Figure 4F, G*). Taken together, these results suggest that fasting-induced hepatic inflammation in *Ripk1*-hepKO mice is characterized by altered hepatic gene expression profiles and changes in liver cell composition, marked by increased recruitment of immune cells to the liver.

To gain further insights, we employed CellChat to analyze cell–cell communication pathways between different cell types, focusing on ligand–receptor pairs (*Jin et al., 2021*). We identified a total of 478 significant ligand–receptor pairs across eight cell types, which were categorized into 53 signaling pathways (*Figure 4—figure supplement 1E, F*). By comparing the information flow of cell–cell communication between *Ripk1*-Con and *Ripk1*-hepKO mice, we found that both the number of ligand–receptor pairs and the interaction strength among the eight cell types were higher in the liver tissue of *Ripk1*-hepKO mice compared to control mice. This was particularly evident in the enhanced crosstalk between macrophages and other cell clusters (*Figure 4H–J*). To analyze the detailed communication within individual pathways, we performed a network analysis that was visualized using a heatmap. This heatmap illustrated the signals between various cell types specifically enriched in *Ripk1*-hepKO mice (*Figure 4K*). These signals were predominantly linked to inflammation-related pathways, such as MHC I, OSM, GAS, and HSPG, as well as proliferation-related pathways including IGF, KIT, NOTCH, and LAMININ. Interestingly, we discovered that fasting amplified SAA signaling in the hepatocytes of *Ripk1*-hepKO mice. This increase in SAA signaling was consistent with elevated transcription levels of SAA2, SAA3, and SAA4, as evidenced by liver transcriptome data (*Figure 3C*). The SAA released by hepatocytes has been reported to play a role in regulating immune responses and tumor development (*Li et al., 2024*; *Stone et al., 2024*). These above results suggest

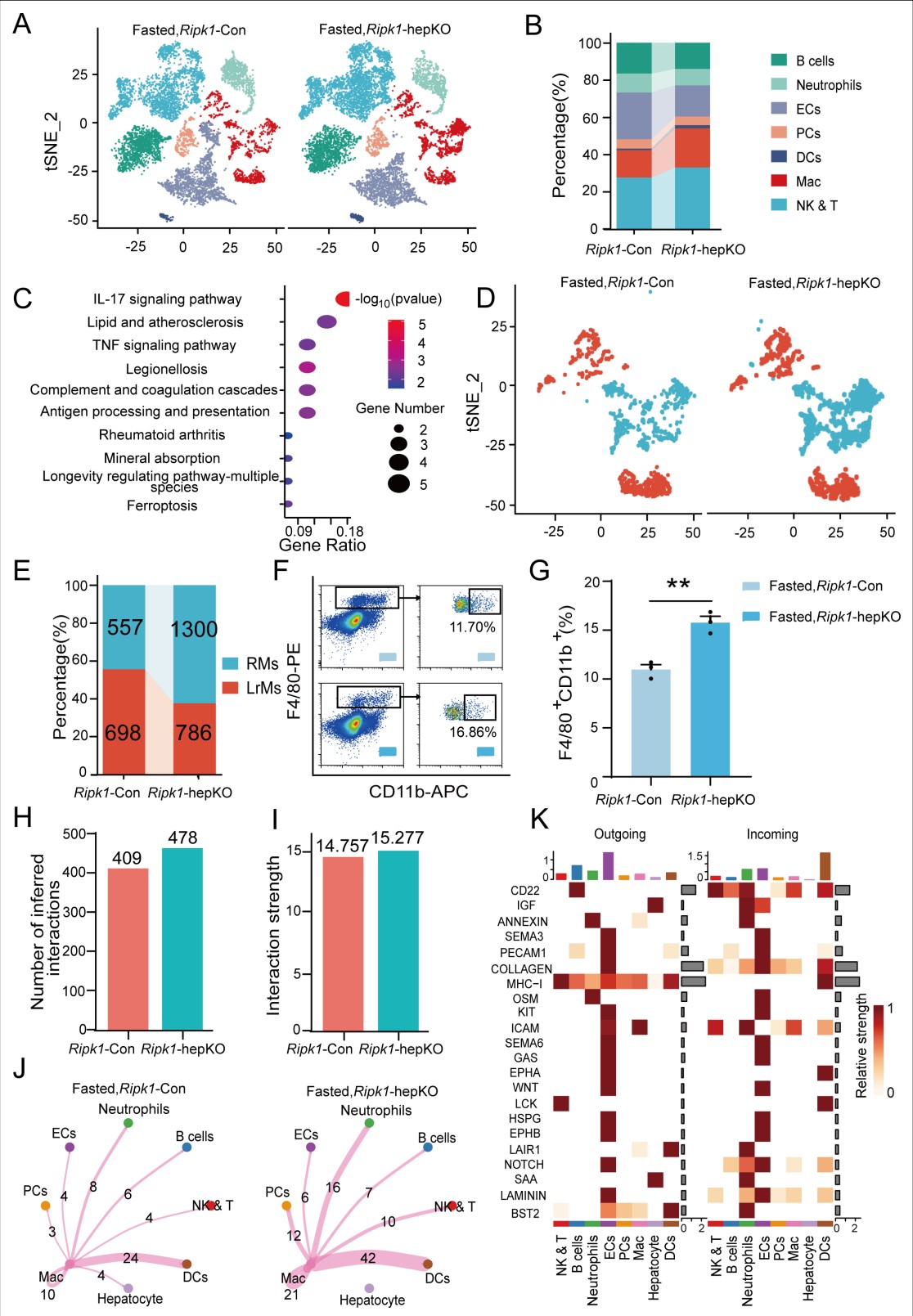

**Figure 4.** Single-cell RNA sequencing of liver tissue from *Ripk1*-Con and *Ripk1*-hepKO mice. (**A**) T-distributed stochastic neighbor embedding (t-SNE) plots display color-coded cell subtypes of cells in the *Ripk1*-Con (left) and *Ripk1*-hepKO (right) mice liver tissues. (**B**) Bar charts display the proportion of major cell subtypes within all different genotypes after fasting. (**C**) The altered signaling pathways associated with differential gene changes in macrophages were enriched by Kyoto Encyclopedia of Genes and Genomes (KEGG) analysis. (**D**) t-SNE plots display color-coded cell subtypes of

*Figure 4 continued on next page*

*Figure 4 continued*

macrophages in the *Ripk1*-Con (left) and *Ripk1*-hepKO (right) mice liver tissues. (**E**) Bar charts display the proportion of major cell subtypes within macrophages after fasting. (**F**) Representative flow cytometry plots of F4/80- and CD11b-positive cells in the liver tissue (*n* = 3 per group). (**G**) Relative increase of F4/80- and CD11b-positive cells in the liver tissue of *Ripk1*-hepKO mice after 12 hr of fasting, compared to that in *Ripk1*-Con mice (*n* = 3 per group; unpaired *t*-test). (**H**) Bar charts display the number of interactions among cell types across the experiments. (**I**) Bar charts displaying the interaction strength among cell types across the experiments. (**J**) Circle plots displaying the brand link pairs between macrophages and other cell types, along with their corresponding event counts. (**K**) Heatmaps summarizing specific signals among interacting cell types in the liver tissue of *Ripk1*-hepKO mice after 12 hr of fasting, compared to that in *Ripk1*-Con mice. Interactions are categorized into outgoing and incoming events for specific cell types. The color gradient indicates the relative strength of these interactions (NK&T: natural killer cells and T cells; ECs: endothelial cells; PCs: plasma cells; Mac: macrophages; DCs: dendritic cells). The data was analyzed via two- or one-way ANOVA. Data are expressed as mean ± SEM (*n* = 6 per group). **p < 0.01.

The online version of this article includes the following figure supplement(s) for figure 4:

**Figure supplement 1.** Single-cell RNA sequencing of liver tissue from *Ripk1*-Con and *Ripk1*-hepKO mice.

that fasting-induced liver injury in RIPK1 knockout mice of hepatic parenchymal cells may exacerbate the inflammatory response in liver tissue through enhanced SAA signaling.

## Endoplasmic reticulum stress is required for fasting-induced liver injury and inflammation in *Ripk1*-hepKO mice

We next aimed to investigate the underlying mechanism of short-term fasting-induced liver injury and inflammation in *Ripk1*-hepKO mice. It is known that fasting-induced lipolysis in peripheral adipose tissue will lead to hepatic lipid burden, and excessive FFA deposition has been shown to induce endoplasmic reticulum stress (ER stress) in liver. In addition, previous studies have shown that RIPK1 promotes apoptosis in response to unresolved ER stress (*Estornes et al., 2014*). As *Figure 1N, O* indicated, the fasting strategy we employed indeed increased the lipid content in liver tissue. We also detected the expression of ER stress markers in the liver tissues of fed- or fasted-mice, respectively. As *Figure 5A* indicated, short-term fasting indeed increased the expression of CHOP, BIP, IRE1a, and phosphor-IRE1a at protein levels, suggesting that short-term fasting caused ER stress in liver. To investigate whether ER stress is involved in the fasting-induced liver injury in *Ripk1*-hepKO mice, *Ripk1*-Con and *Ripk1*-hepKO mice were pretreated with the ER stress inhibitor 4-PBA and then the mice were subjected to short-term fasting for 12 hr (*Figure 5B*). As *Figure 5C* shown, 4-PBA treatment successfully reduced the expression of ER stress markers, confirming the efficiency of 4-PBA to inhibit ER stress. Interestingly, 4-PBA pretreatment effectively reduced serum ALT/AST levels and the number of TUNEL-positive cells in the liver of fasted *Ripk1*-hepKO mice, suggesting that ER stress was involved in fasting-induced liver injury and hepatocyte apoptosis in *Ripk1*-hepKO mice (*Figure 5D–F*). Furthermore, 4-PBA pretreatment also effectively inhibited the increased expression of inflammatory markers (*Ccl2*, *Tnfa*, *Ifng*, and *Il6*) and proliferation markers *Afp* in the liver tissue of fasted *Ripk1*-hepKO mice (*Figure 5G, H*). These results indicated that 4-PBA not only prevented hepatocyte apoptosis and liver injury induced by fasting in *Ripk1*-hepKO mice, but also mitigated the hepatic inflammation and compensatory proliferation, suggesting that ER stress was involved in the process of fasting-induced liver injury and inflammation in *Ripk1*-hepKO mice.

## AAV-TBG-Cre-mediated liver-specific RIPK1 knockout confirms fasting-induced acute liver injury in mice

We employed another different approach to conditionally delete RIPK1 from hepatocytes by injecting AAV8-TBG-Cre virus into *Ripk1*-Con mice. Four weeks after virus injection, the mice were subjected for short-term fasting as above studies, and then serum and liver tissue were harvested (*Figure 6A*). Immunoblot analysis confirmed that RIPK1 was specifically deleted in hepatocytes by AAV8-TBG-Cre virus treatment, in contrast to the control virus AAV8-TBG-null (*Figure 6B*). Our results indicated that AAV8-TBG-Cre virus-induced RIPK1-deficient mice phenocopied *Ripk1*-hepKO mice, that hepatocyte-specific loss of RIPK1 sensitized the mice to short-term fasting-induced acute liver injury and hepatocyte apoptosis (*Figure 6C–F*). Consistently, there was no significant alteration in serum TG, TC, and liver TG levels of *Ripk1*-Con mice injected with AAV8-TBG-Cre or control AAV8-TBG-null virus (*Figure 6G–I*). We also detected the expression of ER stress markers in the liver tissue of AAV8-TBG-null or AAV8-TBG-Cre mice, respectively. As *Figure 6B, J* indicated, short-term fasting indeed

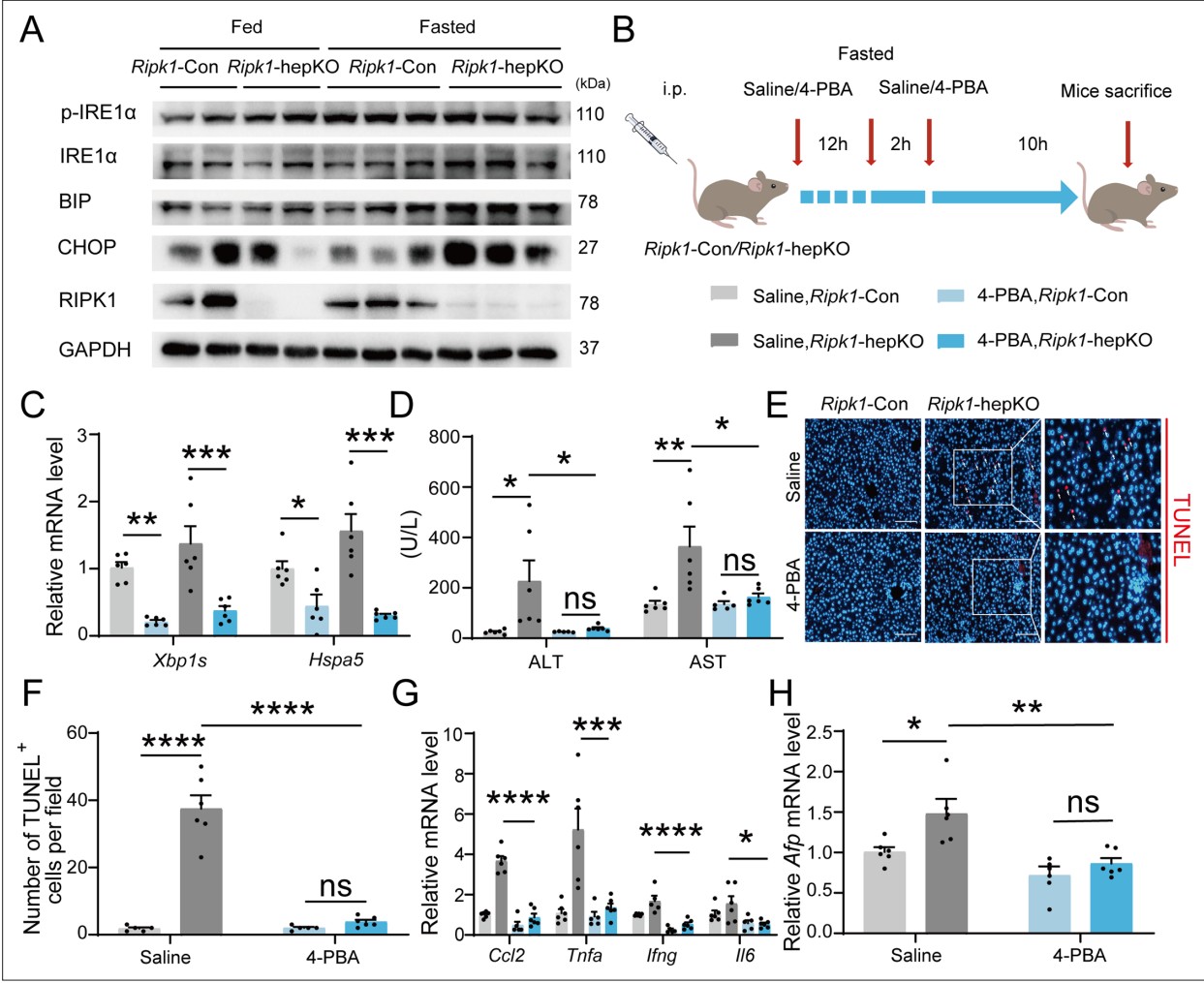

**Figure 5.** Inhibitor of ER stress 4-PBA effectively rescued the fasting-induced liver injury and inflammation in *Ripk1*-hepKO mice. (**A**) Western blot analysis of p-IRE1α, IRE1α, BIP, CHOP, RIPK1, and GAPDH in liver tissue. (**B**) Experiment schema. (**C**) Transcriptional expression of ER stress genes in mouse liver. (**D**) Serum alanine amino-transferase (ALT) and aspartate amino-transferase (AST) levels. (**E, F**) Fluorescence microscopy image and quantification of TUNEL staining. Scale bar, 100 µm. (**G**) Expression (qPCR) of inflammatory genes in the livers. (**H**) Expression (qPCR) of *Afp* in the livers. The data was analyzed via two- or one-way ANOVA. Data are expressed as mean ± SEM (*n* = 6 per group). ns, no significant, *p < 0.05, **p < 0.01, ***p < 0.001, ****p < 0.0001.

The online version of this article includes the following source data for figure 5:

**Source data 1.** PDF file containing original western blots for *Figure 5A*, indicating the relevant bands and treatments.

**Source data 2.** Original files for western blot analysis displayed in *Figure 5A*.

increased the expression of CHOP (*Ddit3*), BIP (*Hspa5*), IRE1α, and phosphor-IRE1α at both mRNA and protein levels, indicating that fasting induced the occurrence of ER stress in RIPK1-deleted livers. As demonstrated in *Figure 6B, K*, transcriptional expression of inflammatory markers, including *Ccl2*, *Tnfa*, and *Il1b*, were also significantly induced in AAV8-TBG-Cre-mouse liver tissue, in contrast to AAV8-TBG-null-mouse. Consistent with previous results, the number of cleaved caspase-3-positive cells was significantly higher in AAV8-TBG-Cre-mouse compared to controls (*Figure 6L, M*). Taken together, these results suggested that hepatocyte-specific loss of RIPK1, achieved by different strategies, made the mice fragile to metabolic disturbance, and even short-term fasting would result in liver injury, hepatocyte cell death, hepatic inflammation, and ER stress.

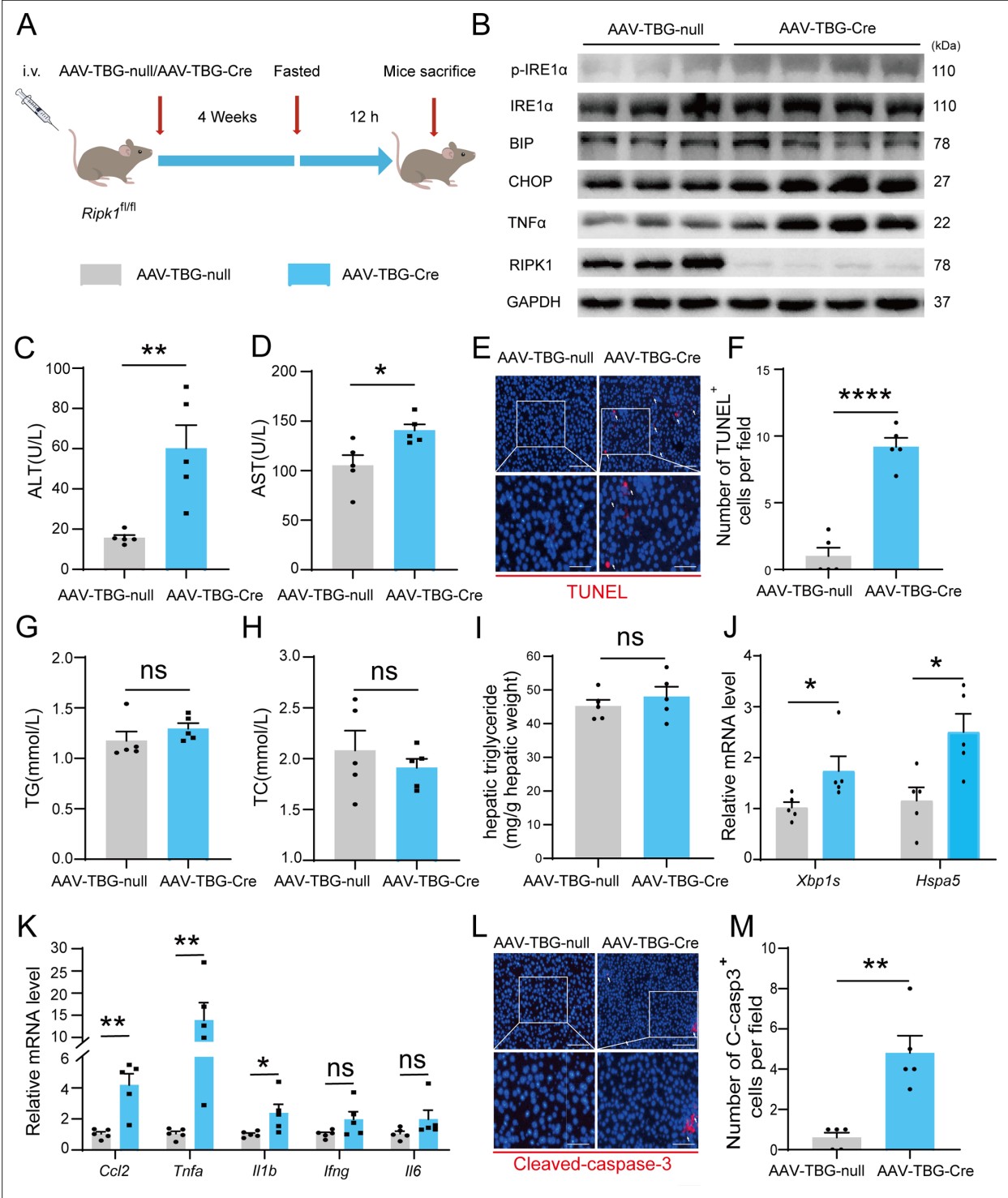

**Figure 6.** AAV-TBG-Cre-mediated liver-specific RIPK1 knockout confirms fasting-induced acute liver injury in mice. (**A**) Schema of AAV8-TBG-Cre administration. (**B**) Western blot analysis of p-IRE1α, IRE1α, BIP, CHOP, TNFα, RIPK1, and GAPDH in liver tissue. (**C**) Serum alanine amino-transferase (ALT) levels. (**D**) Serum aspartate amino-transferase (AST) levels. (**E**, **F**) Fluorescence microscopy image and quantification of TUNEL staining. Scale bar, 100 μm. (**G**) Serum triglycerides (TG) levels. (**H**) Serum total cholesterol (TC) levels. (**I**) Hepatic TG levels (mg/g tissue). (**J**) Expression (qPCR) of ER stress genes in the livers. (**K**) Expression (qPCR) of inflammatory genes in the livers. (**L**, **M**) Fluorescence microscopy images and quantification of liver of *Ripk1*-hepKO and control mice stained with anti-cleaved caspase-3 antibody (red) and DAPI (blue). Scale bar, 100 μm. The data was analyzed via two- or one-way ANOVA. Data are expressed as mean ± SEM (*n* = 6 per group). ns, no significant, *p < 0.05, **p < 0.01, ****p < 0.0001.

The online version of this article includes the following source data for figure 6:

*Figure 6 continued on next page*

*Figure 6 continued*

**Source data 1.** PDF file containing original western blots for *Figure 6B*, indicating the relevant bands and treatments.

**Source data 2.** Original files for western blot analysis displayed in *Figure 6B*.

## Short-term high-fat diet feeding induced liver injury, hepatic apoptosis, inflammation, and ER stress in *Ripk1*-hepKO mice

We next aimed to explore whether other metabolic disturbance exhibited similar effect as short-term fasting in *Ripk1*-hepKO mice. It is known that short-term fasting leads to lipolysis in adipose tissue and temporary lipid burden to hepatocytes. We therefore created a short-term high-fat diet (HFD)

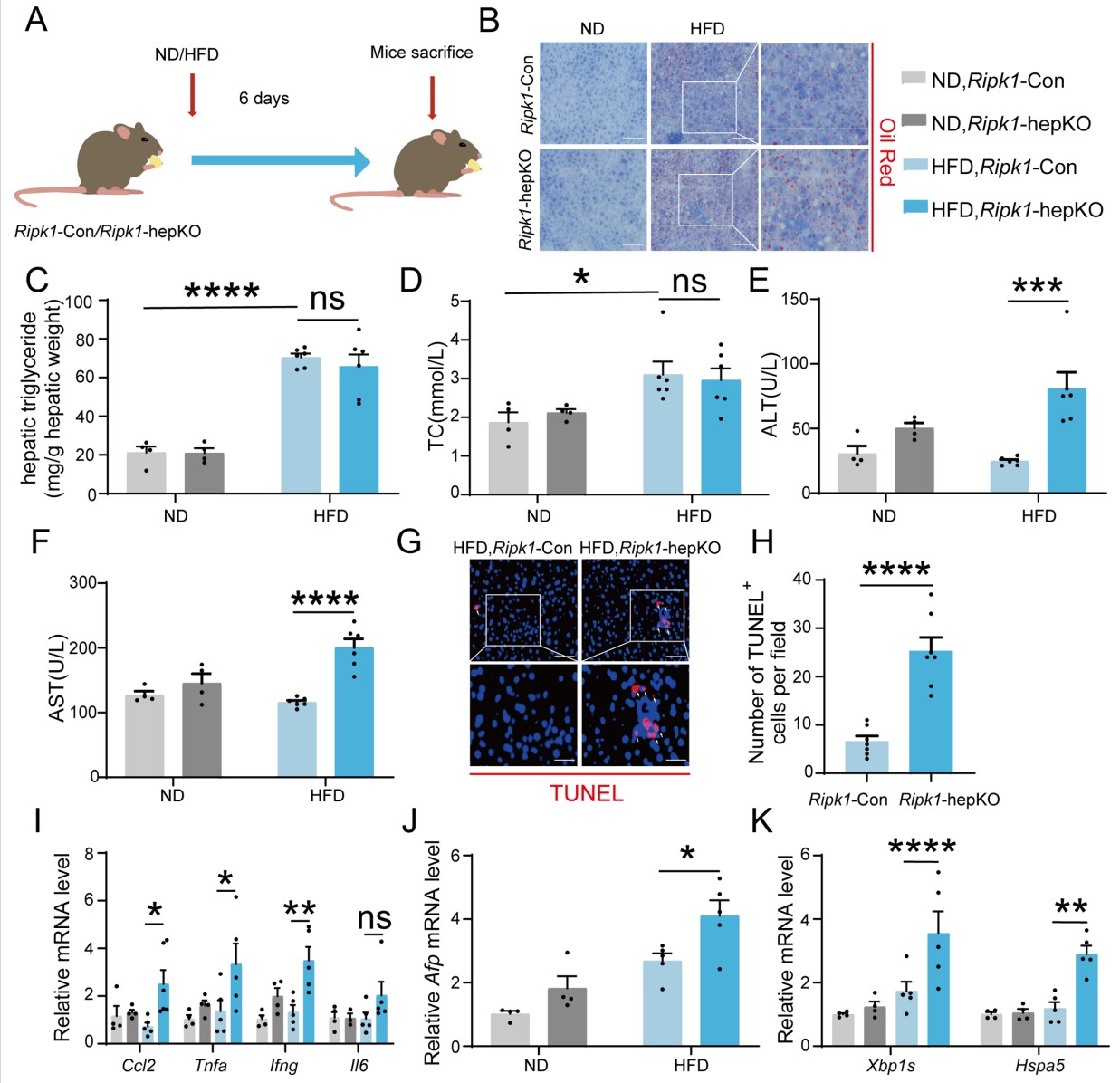

**Figure 7.** Short-term high-fat diet (HFD) feeding induced liver injury, hepatic apoptosis, inflammation, and endoplasmic reticulum stress in *Ripk1*-hepKO mice. (**A**) Schema of HFD administration. (**B**) Liver tissue was stained by Oil Red O. Scale bar, 100 μm. (**C**) Hepatic triglyceride (TG) levels (mg/g tissue). (**D**) Serum total cholesterol (TC) levels. (**E**) Serum alanine amino-transferase (ALT) levels. (**F**) Serum aspartate amino-transferase (AST) levels. (**G**, **H**) Fluorescence microscopy image and quantification of TUNEL staining. Scale bar, 50 μm. (**I**) Expression (qPCR) of inflammatory genes in the livers. (**J**) Expression (qPCR) of *Afp* in the livers. (**K**) Expression (qPCR) of ER stress markers in the livers. The data was analyzed via two- or one-way ANOVA. Data are expressed as mean ± SEM (*n* = 6 per group). ns, no significant, *p < 0.05, **p < 0.01, ****p < 0.0001.

feeding model in both wild-type control and *Ripk1*-hepKO mice, to mimic the dietary-induced lipid metabolism disturbance. *Ripk1*-Con and *Ripk1*-hepKO mice were fed with HFD or normal diet (ND) for 6 days, respectively (*Figure 7A*). We observed a significant increase in liver lipid deposition, liver TG, and serum TC in mice fed with HFD compared to those fed with ND (*Figure 7B–D*). Previous studies reported that short-term HFD feeding was not enough to induce hepatic disorder, without obvious effects on the serum ALT/AST, liver histology, and our results with *Ripk1*-Con mice were in agreement with these findings (*Figure 7E, F*). In contrast, upon 6-day HFD feeding, the serum ALT/AST levels, hepatocyte apoptosis as measured by TUNEL and hepatic inflammation were significantly elevated in *Ripk1*-hepKO mice (*Figure 7E–I*). Additionally, HFD feeding significantly induced the expression of *Afp* in *Ripk1*-hepKO mice, suggesting that HFD feeding promoted tumorigenesis in *Ripk1*-hepKO mice (*Figure 7J*). We also found that the expression of ER stress-related genes was obviously increased in *Ripk1*-hepKO mice after 6 days of HFD feeding (*Figure 7K*).

Collectively, these results suggest that hepatocyte RIPK1 is essential for the maintenance of hepatic homeostasis against the disturbance induced by different metabolism behaviors, including fasting and temporary HFD feeding.

## Discussion

In this study, we demonstrated that the specific deletion of RIPK1 in hepatocytes exacerbated the liver's vulnerability to metabolic disturbances, such as short-term fasting and HFD feeding, resulting in increased liver damage, apoptosis, inflammation, and compensatory proliferation. We utilized single-cell RNA sequencing and bulk RNA sequencing to characterize the hepatic cellular profiles and transcriptional profiles in response to the physiological inflammation induced by the disruption of homeostasis. Furthermore, we presented evidence indicating the involvement of ER stress in sensitizing RIPK1-deleted liver tissue. In summary, we revealed a novel physiological role of RIPK1 as a scaffold in maintaining liver homeostasis during fasting and other nutritional disturbance. These findings could prove valuable in tailoring intermittent fasting or calorie restriction regimens for specific populations based on their *Ripk1* gene polymorphism or expression profiles.

Our work shed light on the intricate interplay between cell death, inflammation, and metabolism. While the pathophysiological roles of RIPK1 have primarily been studied in inflammatory diseases and pathogen infections, emerging evidence suggests its involvement in metabolism-related pathways (*Mei et al., 2021*; *Najafov et al., 2021*; *Zhang et al., 2023a*). Previous studies, including our own, have linked RIPK1's kinase activity to the pathogenesis of metabolic diseases like NASH, indicating a metabolic regulatory role of RIPK1 kinase (*Majdi et al., 2020*; *Tao et al., 2021*; *Yan et al., 2022*). Subsequently, UDP-glucose 6-dehydrogenase and UDP-glucuronate metabolism are found to suppress NASH pathogenesis and control hepatocyte apoptosis through inhibiting RIPK1 kinase activity, further solidifying the connection between RIPK1 kinase activity and metabolism during the pathogenesis of NASH (*Zhang et al., 2023b*). In addition to its pathological roles, RIPK1's involvement in metabolism physiology has become increasingly apparent. Mei et al. found that the postnatal lethality of *Ripk1*$^{-/-}$ mice was attributed to dysregulated aspartate metabolism, leading to impaired starvation-induced autophagy (*Mei et al., 2021*). Zhang et al. recently reported that the classical energy sensor AMPK is able to phosphorylate RIPK1 at S416 in response to metabolic stress like glucose deprivation in vitro and fasting in vivo, and this phosphorylation of RIPK1 by AMPK represents a survival mechanism to keep the kinase activity of RIPK1 in check to prevent RIPK1 kinase-mediated cell death. AMPK deficiency sensitized cells to glucose deprivation-induced cell death (*Zhang et al., 2023a*). Their results directly linked RIPK1 to key metabolism regulator AMPK. Together with our results, it suggested that hepatic AMPK–RIPK1 axis function as a mechanism to maintain liver homeostasis during suffering metabolic stress. Moreover, *Tak1*$^{-/-}$ mice has been shown to exhibit similar phenotypes, suggesting that the key components in the complex I faction in the scaffold complex, all contribute to maintain tissue homeostasis (*Inokuchi-Shimizu et al., 2014*). These findings, together with our study, emphasize the critical role of RIPK1's scaffold function in sensing nutrient stress and maintaining metabolic homeostasis during various starvation conditions, both in neonatal stages and adulthood. Zhang et al. also noted that RIPK1 knockout MEFs (mouse embryonic fibroblasts) were more susceptible to cell death induced by glucose starvation compared to WT MEFs, further underscoring the importance of RIPK1 in nutrient stress responses.

Our study also posited that hepatocyte RIPK1 plays a crucial role in preventing liver damage and inflammation triggered by metabolic stress. Prior investigations have indicated that deficiencies in TAK1 or NEMO in hepatocytes result in spontaneous liver injury and carcinogenesis in mice, with a more severe phenotype than that observed in RIPK1-deficient mice (*Bettermann et al., 2010*; *Luedde et al., 2007*; *Yang et al., 2013*). This is attributed to the deletion of TAK1 or NEMO unleashing the kinase activity of RIPK1, leading to the activation of RIPK1 kinase and the associated cell death pathways. Our findings, combined with these earlier studies, suggested that the RIPK1–TAK1–NF-κB axis constitutes an essential scaffold platform necessary for the liver's adaptation to metabolic fluctuations. Any improper inactivation or deletion of any component within this scaffold axis disrupts the delicate balance between cell death, inflammation, and normal function, rendering the liver vulnerable to metabolic changes and resulting in liver damage, hepatic inflammation, and compensatory proliferation.

Regarding the upstream signal of RIPK1, both short-term fasting and HFD can increase FFA in the bloodstream, leading to their influx and accumulation in the liver. This accumulation may cause lipotoxicity in hepatocytes through ER stress (*Geng et al., 2021*; *Piccolis et al., 2019*). Thus, we hypothesize that lipotoxic stress might result in hepatocyte cell death. We observed that treatment with palmitic acid led to a higher rate of apoptosis in *Ripk1*$^{-/-}$ AML12 liver cells compared to wild-type control cells (data not shown). Moreover, in contrast to organs such as the small intestine and lungs, the liver typically maintains immune tolerance and does not incite inflammation in response to various endogenous and exogenous PAMPs or antigens present in the bloodstream. Recent years have witnessed the increasing attention toward physiological inflammation, which implications have been greatly broadened (*Medzhitov, 2021*; *Meizlish et al., 2021*). Our research revealed that hepatocyte RIPK1 serves as a critical mechanism for preserving immune tolerance within the liver microenvironment. Furthermore, we identify an instance of physiological inflammation induced by loss of regulation in normal tissue. The physiological inflammation observed in RIPK1-deficient livers during short-term fasting is milder but akin to hepatic inflammation induced by pathogenic infections and other pathological conditions. It is characterized by the upregulation of typical molecules, including *Ccl2*, *Tnfa*, *Ifng*, *Il6*, etc., and the recruitment of macrophages into liver tissue, indicating systemic adaptive responses.

In summary, our study revealed the multifaceted role of RIPK1 in maintaining liver homeostasis in the face of metabolic challenges, shedding light on the intricate interplay between cell death, inflammation, and metabolism. Our findings provide new insights into the role of RIPK1 in various physiological contexts.

# Materials and methods

## Key resources table

| Reagent type (species) or resource | Designation | Source or reference | Identifiers | Additional information |
|---|---|---|---|---|
| Commercial assay or kit | Alanine aminotransferase Assay Kit | Nanjing Jiancheng Bioengineering Institute | C009-2-1 | |
| Commercial assay or kit | Aspartate aminotransferase Assay Kit | Nanjing Jiancheng Bioengineering Institute | C010-2-1 | |
| Commercial assay or kit | Triglyceride assay kit | Nanjing Jiancheng Bioengineering Institute | A110-1-1 | |
| Commercial assay or kit | Total cholesterol assay kit | Nanjing Jiancheng Bioengineering Institute | A111-1-1 | |
| Commercial assay or kit | Free fatty acids (FFA) Content Assay Kit | Solarbio | BC0595 | |
| Commercial assay or kit | Ketone Body Assay | Abcam | ab272541 | |
| Commercial assay or kit | Triglyceride (TG) Content Assay Kit | Solarbio | BC0625 | |
| Commercial assay or kit | TUNEL BrightRed Apoptosis Detection Kit | Vazyme | A113-02 | |

*Continued on next page*

*Continued*

| Reagent type (species) or resource | Designation | Source or reference | Identifiers | Additional information |
|---|---|---|---|---|
| Commercial assay or kit | Fixation/Permeabilization Buffer Set | ABclonal | RK50005 | |
| Antibody | anti-Caspase-3 (Rabbit polyclonal) | Cell Signaling Technology | Cat#: 9662S, RRID:AB_331439 | WB (1:1000) |
| Antibody | anti-MLKL (Rabbit monoclonal) | Cell Signaling Technology | Cat#: 37705S, RRID:AB_2799118 | WB (1:1000) |
| Antibody | anti-cleaved-Caspase-3 (Rabbit polyclonal) | Cell Signaling Technology | Cat#: 9661S, RRID:AB_2341188 | IF (1:400), WB (1:1000) |
| Antibody | anti-phospho-MLKL (S345) (Rabbit monoclonal) | Cell Signaling Technology | Cat#: 37333S, RRID:AB_2799112 | IF (1:1600), WB (1:1000) |
| Antibody | anti-F4/80 (Rabbit polyclonal) | Servicebio | Cat#: GB113373, RRID:AB_2938980 | IF (1:750) |
| Antibody | anti-CD11b (Rabbit polyclonal) | Servicebio | Cat#: GB11058, RRID:AB_2928112 | IF (1:750) |
| Antibody | anti-CCR2 (Rabbit polyclonal) | Servicebio | Cat#: GB11326 | IF (1:2000) |
| Antibody | anti-CX3CR1 (Rabbit polyclonal) | Servicebio | Cat#: GB11861 | IF (1:2500) |
| Antibody | anti-MKI67 (Rabbit monoclonal) | Bioss | Cat#: bs-2130R, RRID:AB_10857118 | IHC (1:500) |
| Antibody | anti-ALB (Mouse monoclonal) | Servicebio | Cat#: GB122080 | IF (1:1500) |
| Antibody | anti-TNF-alpha (Mouse monoclonal) | proteintech | Cat#: 60291-1-Ig, RRID:AB_2833255 | IF (1:250), WB (1:2500) |
| Antibody | anti-RIPK1 (Rabbit monoclonal) | Cell Signaling Technology | Cat#: 3493S, RRID:AB_2305314 | WB (1:1000) |
| Antibody | anti-phospho-IRE1α (S724) (Rabbit polyclonal) | Abcam | Cat#: ab48187, RRID:AB_873899 | WB (1:1000) |
| Antibody | anti-IRE1α (Rabbit monoclonal) | Cell Signaling Technology | Cat#: 3294S, RRID:AB_823545 | WB (1:1000) |
| Antibody | anti-BIP (Rabbit monoclonal) | Cell Signaling Technology | Cat#: 3177S, RRID:AB_2119845 | WB (1:1000) |
| Antibody | anti-CHOP (Mouse monoclonal) | Cell Signaling Technology | Cat#: 2895S, RRID:AB_2089254 | WB (1:1000) |
| Antibody | anti-CD11b-ABflo 647 (Rabbit monoclonal) | ABclonal | Cat#: A24095 | FACS (5 µl per test) |
| Antibody | anti-F4/80-PE (Rabbit monoclonal) | ABclonal | Cat#: A25659 | FACS (5 µl per test) |
| Antibody | anti-Ki67-ABflo 488 (Rabbit monoclonal) | ABclonal | Cat#: A26239 | FACS (5 µl per test) |
| Antibody | anti-beta Actin (Mouse monoclonal) | Servicebio | Cat#: GB12001, RRID:AB_2904016 | WB (1:1000) |
| Antibody | anti-GAPDH (Mouse monoclonal) | Servicebio | Cat#: GB12002, RRID:AB_3206256 | WB (1:1000) |
| Sequence-based reagent | *Ccl2*_F | This paper | PCR primers | CACTCACCTGCTGCTACTCA |
| Sequence-based reagent | *Ccl2*_R | This paper | PCR primers | AGACCTTAGGGCAGATGCAG |
| Sequence-based reagent | *Tnfa*_F | This paper | PCR primers | GTAGCCCACGTCGTAGCAAA |
| Sequence-based reagent | *Tnfa*_R | This paper | PCR primers | TAGCAAATCGGCTGACGGTG |
| Sequence-based reagent | *Il1b*_F | This paper | PCR primers | GCCACCTTTTGACAGTGATGAG |
| Sequence-based reagent | *Il1b*_R | This paper | PCR primers | ACGGGAAAGACACAGGTAGC |
| Sequence-based reagent | *Ifng*_F | This paper | PCR primers | ATGAACGCTACACACTGCATC |

*Continued on next page*

*Continued*

| Reagent type (species) or resource | Designation | Source or reference | Identifiers | Additional information |
|---|---|---|---|---|
| Sequence-based reagent | *Ifng*_R | This paper | PCR primers | CCATCCTTTTGCCAGTTCCTC |
| Sequence-based reagent | *Il6*_F | This paper | PCR primers | TGATGGATGCTACCAAACTGGA |
| Sequence-based reagent | *Il6*_R | This paper | PCR primers | GGAAATTGGGGTAGGAAGGACT |
| Sequence-based reagent | *Afp*_F | This paper | PCR primers | ACCTCCAGGCAACAACCATT |
| Sequence-based reagent | *Afp*_R | This paper | PCR primers | GTTTGACGCCATTCTCTGCG |
| Sequence-based reagent | *Mki67*_F | This paper | PCR primers | AACCATCATTGACCGCTCCT |
| Sequence-based reagent | *Mki67*_R | This paper | PCR primers | AGGCCCTTGGCATACACAAA |
| Sequence-based reagent | *Hspa5*_F | This paper | PCR primers | CGAGGAGGAGGACAAGAAGG |
| Sequence-based reagent | *Hspa5*_R | This paper | PCR primers | TCAAGAACGGGCAAGTTCCAC |
| Sequence-based reagent | *Xbp1s*_F | This paper | PCR primers | GAGTCCGCAGCAGGTG |
| Sequence-based reagent | *Xbp1s*_R | This paper | PCR primers | AGGGTACCTGAGACTGTG |
| Sequence-based reagent | *Ddit3*_F | This paper | PCR primers | CTGCCTTTCACCTTGGAGAC |
| Sequence-based reagent | *Ddit3*_R | This paper | PCR primers | ATAGAGTAGGGGTCCTTTGC |
| Sequence-based reagent | *Gapdh*_F | This paper | PCR primers | AGGTCGGTGTGAACGGATTTG |
| Sequence-based reagent | *Gapdh*_R | This paper | PCR primers | TGTAGACCATGTAGTTGAGGTCA |

## Mouse models

*Ripk1*$^{flox/flox}$ (*Ripk1*-Con) and *Albumin*-cre transgenic mice on C57BL/6J background as previously described were kindly provided by Prof. Haibing Zhang (Chinese Academy of Sciences, Shanghai, China), both male and female littermates were used in the experiments (*Schneider et al., 2017*). Liver-specific RIPK1 knockout mice (*Ripk1*-hepKO mice) were generated by crossing *Ripk1*-Con and *Albumin*-cre mice. Then *Ripk1*-Con mice were bred with *Ripk1*-Con$^{Cre/+}$ mice to generate *Ripk1*-Con and *Ripk1*-Con$^{Cre/+}$ mice as littermates. All mice were maintained in the standard laboratory conditions with temperature 22 ± 2°C, strict 12-hr dark–light cycles (7:00 a.m. to 7:00 p.m.). All mice had ad libitum access to food and water, and before the start of experiments all mice were maintained on a regular chow diet (LAD 0011, Trophic Animal Feed, China). All experiments were conducted following the Animal Care and Use Committee at Nanjing University of Science & Technology (AUCU-NUST-20210287052).

For the fasting studies involving *Ripk1*-Con and *Ripk1*-hepKO mice, 6-week-old littermates of *Ripk1*-Con or *Ripk1*-hepKO were randomly assigned to groups. The fed group was allowed ad libitum access to both food and water. The fasted group, on the other hand, was deprived of food but not water from 8:00 p.m. since then, and this deprivation continued for a period of 12 hr.

For the ER stress inhibitor experiment, 6-week-old *Ripk1*-Con and *Ripk1*-hepKO mice were divided into separate groups and administered either 80 mg/kg of 4-PBA (SML0309, Sigma, Germany) or saline vehicle via intraperitoneal injection at 8:00 a.m. Fasting was initiated for all groups by removing food at 8:00 p.m., while water was still available, for a duration of 12 hr. At 10:00 p.m., the mice were intraperitoneally injected with 80 mg/kg of 4-PBA.

For the HFD experiment, separate groups of *Ripk1*-Con and *Ripk1*-hepKO mice were fed either an HFD (TP201482; Trophic Animal Feed, China) or an ND (LAD 0011, Trophic Animal Feed, China) for a period of 6 days. On the last day, a 3-hr fast was initiated at 7 a.m. and the mice were sacrificed thereafter.

For the adenovirus-associated virus (AAV) experiment, 6-week-old littermates of *Ripk1*-Con were randomly divided into two groups. One group was administered serotype 8 AAV-thyroxine-binding globulin (TBG)-Cre (AAV-TBG-Cre, $1.5 \times 10^{11}$ gene copies per mouse, intravenously) (H5721, Obio Technology, China), while the other group was injected with AAV-TBG-null (H20594, Obio Technology, China). After 4 weeks, the mice were subjected to the same fasting rationale as described earlier.

At the end of the experiment of each different model, mice were sacrificed following blood collection. Subsequently, the mice were dissected and the liver tissues were harvested. Some portions of the tissues were fixed in 4% paraformaldehyde tissue fixation solution for further histological examination, while others were stored at −80°C for additional detection. All mice used for experiments displayed general health. The total number of mice analyzed for each experiment is detailed in the figure legends.

## Serum and liver assays

Plasma was maintained at room temperature for 1 hr, and then centrifuged at 4000 rpm for 10 min to take the supernatant as serum. Serum alanine transaminase (ALT), aspartate transaminase (AST), TG, TC, FFA, and β-hydroxybutyrate were measured according to the manufacturer's instructions. For measurements of liver TG levels, 50 mg liver was homogenized in lysis buffer (isopropanol:chloroform = 1:1) using a high speed low temperature tissue grinding machine (Servicebio, China) and supernatant was centrifuged $8000 \times g$ for 10 min at 4°C for measurement according to the manufacturer's instructions.

## TUNEL assay

TUNEL Assay Kits were used to measure apoptotic cells in livers. Briefly, paraffin-embedded liver tissue sections were dried for 30 min at 60°C, and soaked in xylene to deparaffinized completely. Deparaffinized sections were pretreated by proteinase K for 20 min at 25°C. Then phosphate-buffered saline (PBS) was used to wash tissue sections and the mixture of reaction buffer with TdT enzyme was used to incubate in dark for 60 min at 25°C. Meanwhile the cell nucleuses were stained with DAPI. The tissue sections were observed and photographed using the fluorescence microscope (NIKON ECLIPSE 80i). The number of apoptotic cells was quantified in at least three random fields of the same size.

## RNA isolation and RT-qPCR

Liver tissue (10–20 mg) was homogenized directly in TRIzol (Invitrogen) using a high speed low temperature tissue grinding machine (Servicebio, China), then total mRNA was extracted according to the manufacturer's instructions. Next cDNA was synthesized using one-step reverse transcription kit (5x All-In-One-RT Master Mix, Abm). RT-qPCR reactions were performed on the iQ5 real-time PCR system (Bio-Rad, USA) using Hieff qPCR SYBR Green Master Mix (Yeasen Biotech, China). GAPDH mRNA was used as an internal control to normalize mRNA expression. The sequences of primers for qPCR were as Key Resources Table.

## Histological analysis

Liver tissues were fixed in 4% paraformaldehyde and embedded in paraffin for immunohistochemistry/immunofluorescence. For histopathology, frozen sections of liver tissues were stained by Oil Red O working solution to determine the steatosis in liver tissues.

## Immunohistochemistry and immunofluorescence analysis

For immunohistochemistry in liver tissues, paraffin-embedded mouse liver sections (3.5–4 μm) were dried for 30 min at 60°C, and soaked in xylene to deparaffinized completely. Then boiled the liver sections in citrate buffer (pH = 6.0) for antigen retrieval, followed by hydrogen peroxide blocking of endogenous peroxidase. Then sections were incubated with primary antibody (anti-MKI67) at 4°C overnight. The slides were then washed with PBS (PH = 7.4) and incubated with secondary antibody

for 1 hr at 25°C, and then stained with DAB substrate after 5 min with streptavidin-HRP. The cell nuclei were stained with hematoxylin for 3 min at 25°C. For immunofluorescence, after being incubated with primary antibodies (anti-cleaved-caspase-3, anti-phospho-MLKL (S345), anti-F4/80, anti-CD11b, anti-CCR2, and anti-CX3CR1) at 4°C overnight, slides were washed with PBS (pH = 7.4) for three times and then incubated with fluorescent secondary antibodies in dark for 60 min at 25°C. The cell nuclei were stained in dark by DAPI for 5 min at 25°C. Finally, the slides were observed and photographed using the fluorescence microscope (NIKON ECLIPSE 80i). The number of positive cells was quantified in at least three random fields of the same size.

## Western blotting

Total protein was extracted from liver tissues with RIPA lysis buffer (Beyotime, China) with protease inhibitor (P1005; Beyotime, China) and phosphatase inhibitor (P1081; Beyotime, China). BCA Protein Assays kits (Yeasen, China) were used to measure the protein concentrations. Equal amounts of proteins were separated on sodium dodecyl sulfate–polyacrylamide gel electrophoresis on 4–12% acrylamide gel (Yeasen, China) and transferred onto 0.22 μm PVDF membranes (Millipore Corporation, USA). Membranes were incubated with primary antibodies (anti-RIPK1, anti-MLKL, anti-Caspase-3, anti-TNFα, anti-phospho-IRE1α, anti-IRE1α, anti-BIP, and anti-CHOP) overnight at 4°C, washed with TBS-T (0.1% Tween-20) and incubated with suitable secondary antibodies (Goat anti-Mouse IgG HRP or Goat anti-Rabbit IgG HRP) at room temperature for 1 hr. A chemiluminescent reagent (Beyotime, China) and a ChemiDoc MP Imaging System (CLiNX, China) were used for detection.

## Flow cytometry assay

Single-cell suspensions were prepared from 6-week-old *Ripk1*-Con and *Ripk1*-hepKO mice that had been fasted for 12 hr, utilizing the retrograde perfusion method (*Aparicio-Vergara et al., 2017*). The harvested cells were washed twice with PBS. To assess immune cell recruitment in liver tissue, F4/80 was employed to label immune cells, while CD11b was used to label myeloid cells, following the manufacturer's instructions for flow cytometry. For the detection of MKI67-positive cells, fixation and permeabilization were performed using Fixation/Permeabilization Buffer (ABclonal, China) prior to flow cytometry analysis.

## Liver tissue transcriptome sequencing

### Isolation of RNA for sequencing

Six-week-old *Ripk1*-Con and *Ripk1*-hepKO male mice were fasted for 12 hr. At end of the fast, mice were sacrificed and dissected. Approximately 200 mg of liver tissue at the left lateral lobe was cut, weighed and harvested for each mouse. Total RNA was extracted from mouse liver using TRIzol reagent (Invitorgen) following the manufacturer's instructions. The isolated RNA was further quantified using a NanoDrop 2000 spectrophometer (Thermo Fisher Scientific). Then the total RNA was subjected to sequencing for transcriptome analysis.

### cDNA library construction and sequencing

The RNA-seq library was generated using the TruSeq Stranded Total RNA Sample Preparation kit from Illumina (NEB, USA) following the manufacturer's recommendations for 150 bp paired-end sequencing. Poly-A-containing mRNA were purified and fragmented from DNase-treated total RNA using oligo(dT) magnetic beads. Following the purification step, mRNAs were fragmented to 300 bp nucleotides and used for synthesis of single-stranded cDNAs by means of random hexamer priming. With the constructed single-stranded cDNAs as templates, second strand cDNA synthesis was performed to prepare double-stranded cDNAs. These cDNAs were then amplified with end-pair repair, addition of A-tail, and adapter ligation using polymerase chain reaction. The quality control of cDNA library was evaluated by Agilent 2100 BioAnalyzer (Agilent Technologies). After quality control, sequencing was performed with Illumina HiSeq3000.

### Transcript data analysis

Illumina sequencing analysis viewer was used for data quality check and Illumina Bcl2fastq2 program was used for demultiplexing. Then the high-quality filtered sequencing data was aligned with mouse reference genome (GRCm38) by TopHat (version 2.1.1). To profile DEG, reads per kilobase per million

mapped reads (RKM) were calculated for each transcript. Paired sample *t*-tests followed by Benjamini–Hochberg correction were performed to identify DEGs. Genes with >twofold changes (|log2 fold change|> 1) and false discovery rate <0.05 were defined as DEGs by DESeq2 (version 1.18.1). Enrichment analysis of DEGs was performed by KEGG (http://www.kegg.jp/) pathway enrichment analysis to identify the affected metabolic pathways and signal transduction pathways. The data has been uploaded to the GEO database (GSE286073).

## Single-cell RNA sequencing

Male mice of 6 weeks old with *Ripk1*-Con and *Ripk1*-hepKO genotypes were subjected to a 12-hr fast, after which they were sacrificed and dissected. Liver tissue samples were collected from the left lateral lobe, right lateral lobe, and right medial lobe of each mouse and weighed, with approximately 600 mg of tissue per sample. Liver tissue samples were mechanically dissociated and digested in lysis buffer containing collagenase/dispase and 0.001% deoxyribonuclease I, after which the samples were processed. Isolated cells passed through 70 µm cell strainer were treated with Red Blood Cell Lysis Solution (10×) (Miltenyi Biotec) for 5 min to lyse blood cells. To acquire cells with >90% viability, dead cells were removed with Dead Cell Removal Kit (Miltenyi Biotec) according to the manufacturer's instructions and cell pellet was resuspended in 2% fetal bovine serum/PBS. Then live cells were counted with Countstar and adjusted the cell concentration to 1000 cells/µl. For scRNA-seq, chromium microfluidic chips were loaded with cell suspension with 3′ chemistry and barcoded with a 10× Chromium Controller (10× Genomics). RNA was reverse-transcribed from the barcoded cells. Qubit dsDNA HS Assay Kit (Invitrogen) was used to quantify cDNA concentration and single-cell transcriptome libraries were constructed using the 10× Chromium Single Cell 3′ Library (10× Genomics, v3 barcoding chemistry). Quality control was performed with Agilent 2100 (Agilent Technologies). Libraries were then purified, pooled, and analyzed on Illumina NovaSeq 6000 S2 Sequencing System with 150 bp paired-end reads according to the manufacturer's instructions.

For analysis of scRNA-seq data, briefly, cellRanger single-cell software suite (v3.1.0) were used to process and analysis scRNA-seq reads with the default parameters. Base calling files generated by Illumina sequencer were demultiplexed according to the sample index. Sequences were then aligned to mouse reference genome mm10 reference for whole transcriptome analysis. CellRanger 3.1.0 were used to conducted Filtering, barcode counting, and UMI counting. For the following analysis multiple samples were aggregated. Visualization, quality control, normalization, scaling, PCA dimension reduction, clustering, and differential expression analysis were used by Loupe browser and Seurat (v4.0.0) to perform. Cells with a detected gene number of <200 or >10,000, mitochondrion gene percentage of <10, hemoglobin gene percentage of <10, and doublet nucleus were removed by package Seurat (v4.0.0). Data were subsequently log-normalized (divided by the total expression and amplified scaling factor 10,000) before further analyses. The remaining 22,274 cells were unsupervised clustered after aligning the top 30 dimensions and setting resolution to 0.9. Loupe browser software (10× Genomics) was used to demonstrate UMAP and t-SNE. The identity for each cluster was assigned according to marker genes for known cell types in the mouse liver. DEGs with absolute log-fold change greater than 1.5 and p value less than 0.05 were used for pathway and network enrichment analysis on the KEGG website (http://www.kegg.jp/). The R package CellChat was used to infer potential intercellular communication among different major cellular clusters (*Jin et al., 2021*). We first identified significant ligand–receptor pairs that mediate signaling. CellChat then analyzed the information flows and revealed both incoming and outgoing communication patterns for selected cells. To compare cellular interactions across different samples, we conducted separate analyses on *Ripk1*-Con and *Ripk1*-hepKO mice liver samples. The inferred cellular interactions were then merged to visualize and highlight the differences between the two samples. The data has been uploaded to the GEO database (GSE286072).

## Statistical analysis

All the results in this study are expressed as mean ± SEM. Unpaired Student's *t*-test or one-/two-way ANOVA (for more than two groups) analysis was used to calculate the differences in mean values, using GraphPad Prism 8.2.1 software (San Diego, USA). p < 0.05 was considered as statistically significant.

## Acknowledgements

This study was supported by the Noncommunicable Chronic Diseases-National Science and Technology Major Project (2023ZD0508800), the National Natural Science Foundation of China under Grant 22376100 and 31970897, and Chengdu Medical Research Project 2024242.

## Additional information

### Funding

| Funder | Grant reference number | Author |
| --- | --- | --- |
| National Science and Technology Major Project | 2023ZD0508802 | Dan Weng |
| National Natural Science Foundation of China | 22376100 | Dan Weng |
| National Natural Science Foundation of China | 31970897 | Dan Weng |

The funders had no role in study design, data collection and interpretation, or the decision to submit the work for publication.

### Author contributions

Weigao Zhang, Conceptualization, Data curation, Validation, Investigation, Visualization, Methodology, Writing – original draft, Project administration, Writing – review and editing; Hu Liu, Validation, Investigation, Visualization, Methodology, Writing – review and editing; Danyang Zhang, Data curation, Validation, Investigation, Visualization, Methodology, Writing – original draft; Yuguo Yi, Conceptualization, Formal analysis, Investigation, Visualization, Methodology; Liang Tao, Conceptualization, Data curation, Investigation, Visualization, Methodology; Yunfeng Zhu, Shuxian Huang, Xunan Zhao, Qianchao Shao, Peiqi Li, Yiwen Weng, Investigation, Visualization, Methodology; Wei Lu, Supervision, Investigation, Visualization, Methodology; Jianfa Zhang, Conceptualization, Resources, Methodology; Haibing Zhang, Conceptualization, Resources, Data curation, Formal analysis, Supervision, Validation, Investigation, Methodology, Project administration; Yuxin Chen, Conceptualization, Resources, Data curation, Formal analysis, Supervision, Validation, Investigation, Methodology, Writing – original draft, Project administration, Writing – review and editing; Dan Weng, Conceptualization, Resources, Data curation, Formal analysis, Supervision, Funding acquisition, Investigation, Visualization, Methodology, Writing – original draft, Project administration, Writing – review and editing

### Author ORCIDs

Jianfa Zhang ® https://orcid.org/0000-0002-8814-9103
Dan Weng ® https://orcid.org/0000-0001-8078-6864

### Ethics

All experiments were conducted following the Animal Care and Use Committee at Nanjing University of Science & Technology (AUCU-NUST-20210287052).

Reviewer #1 (Public review): https://doi.org/10.7554/eLife.96798.3.sa1
Reviewer #2 (Public review): https://doi.org/10.7554/eLife.96798.3.sa2
Author response https://doi.org/10.7554/eLife.96798.3.sa3

## Additional files

### Supplementary files

MDAR checklist

### Data availability

Sequencing data have been deposited in GEO under accession codes GSE286072 and GSE286073.

The following datasets were generated:

| Author(s) | Year | Dataset title | Dataset URL | Database and Identifier |
|---|---|---|---|---|
| Zhang W, Liu H, Zhang D, Yi Y, Tao L, Zhang H, Chen Y, Weng D | 2025 | Role of Hepatocyte RIPK1 in Maintaining Liver Homeostasis during Metabolic Challenges [scRNA-seq] | http://www.ncbi.nlm.nih.gov/geo/query/acc.cgi?acc=GSE286072 | NCBI Gene Expression Omnibus, GSE286072 |
| Zhang W, Liu H, Zhang D, Yi Y, Tao L, Zhang H, Chen Y, Weng D | 2025 | Role of Hepatocyte RIPK1 in Maintaining Liver Homeostasis during Metabolic Challenges [RNA-seq] | http://www.ncbi.nlm.nih.gov/geo/query/acc.cgi?acc=GSE286073 | NCBI Gene Expression Omnibus, GSE286073 |

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
