## [Editor Report · eLife Assessment]

This **important** study provides insights into the physiological role of RIPK1 in liver physiology, particularly during short-term fasting. The discovery that RIPK1 deficiency sensitizes the liver to acute injury and hepatocyte apoptosis is based on **convincing** evidence, highlighting the importance of RIPK1 in maintaining liver homeostasis under metabolic stress. The work will be of relevance to anyone studying liver pathologies.

---

## [Referee Report · Reviewer #1 (Public review)]

This study presents an investigation into the physiological functions of RIPK1 within the context of liver physiology, particularly during short-term fasting. Through the use of hepatocyte-specific Ripk1-deficient mice (Ripk1Δhep), the authors embarked on an examination of the consequences of Ripk1 deficiency in hepatocytes under fasting conditions. They discovered that the absence of RIPK1 sensitized the liver to acute injury and hepatocyte apoptosis during fasting, a finding of significant interest given the crucial role of the liver in metabolic adaptation. Employing a combination of transcriptomic profiling and single-cell RNA sequencing techniques, the authors uncovered intricate molecular mechanisms underlying the exacerbated proinflammatory response observed in Ripk1Δhep mice during fasting. While the investigation offers valuable insights into the consequences of Ripk1 deficiency in hepatocytes during fasting conditions, there appears to be a primarily descriptive nature to the study with a lack of clear connection between the experiments. Thus, a stronger focus is warranted, particularly on understanding the dialogue between hepatocytes and macrophages. Moreover, the data would benefit from reinforcement through additional experiments such as Western blotting, flow cytometry, and rescue experiments, which would offer a more quantitative aspect to the findings. By incorporating these enhancements, the study could achieve a more comprehensive understanding of the underlying mechanisms and ultimately strengthen the overall impact of the research.

Comments on revision:

The authors have addressed my comments accordingly.

---

## [Referee Report · Reviewer #2 (Public review)]

Summary:

Zhang et al. analyzed the functional role of hepatocyte RIPK1 during metabolic stress, particularly its scaffold function rather than kinase function. They show that Ripk1 knockout sensitizes the liver to cell death and inflammation in response to short-term fasting, a condition that would not induce obvious abnormality in wild-type mice.

Strengths:

The findings are based on a knockout mouse model and supported by bulk RNA-seq and scRNA-seq. The work consolidates the complex role of RIPK1 in metabolic stress.

Comments on revision:

The authors have addressed my concerns. The added experiments consolidated the findings. I do not have further comments.

---

## [Author Response]

The following is the authors’ response to the original reviews.

**eLife Assessment**
The study presents valuable findings on the role of RIPK1 in maintaining liver homeostasis under metabolic stress. Strengths include the intriguing findings that RIPK1 deficiency sensitizes the liver to acute liver injury and apoptosis, but because the conclusions require additional experimental support, the evidence is incomplete.

We are truly grateful, and wish to express our sincere acknowledgement to the reviewer and the editor for the time and effort spent in reviewing our manuscript. We highly appreciate the thorough and constructive comments, which can greatly improve our manuscript. We have conducted new experiments to address the reviewer’s concerns. We also carefully checked and changed our manuscript according to the constructive suggestions by the reviewer. Hopefully we have adequately addressed all the concerns. In the revised manuscript version, changes are highlighted in yellow. Please find the detailed point-to-point responses below.

**Public Reviews:**

**Reviewer #1 (Public Review):**
This study presents an investigation into the physiological functions of RIPK1 within the context of liver physiology, particularly during short-term fasting. Through the use of hepatocyte-specific Ripk1-deficient mice (Ripk1Δhep), the authors embarked on an examination of the consequences of Ripk1 deficiency in hepatocytes under fasting conditions. They discovered that the absence of RIPK1 sensitized the liver to acute injury and hepatocyte apoptosis during fasting, a finding of significant interest given the crucial role of the liver in metabolic adaptation. Employing a combination of transcriptomic profiling and single-cell RNA sequencing techniques, the authors uncovered intricate molecular mechanisms underlying the exacerbated proinflammatory response observed in Ripk1Δhep mice during fasting. While the investigation offers valuable insights into the consequences of Ripk1 deficiency in hepatocytes during fasting conditions, there appears to be a primarily descriptive nature to the study with a lack of clear connection between the experiments. Thus, a stronger focus is warranted, particularly on understanding the dialogue between hepatocytes and macrophages. Moreover, the data would benefit from reinforcement through additional experiments such as Western blotting, flow cytometry, and rescue experiments, which would offer a more quantitative aspect to the findings. By incorporating these enhancements, the study could achieve a more comprehensive understanding of the underlying mechanisms and ultimately strengthen the overall impact of the research.

We thank the reviewer for the encouraging comments and helpful suggestions. We agree with the reviewer that additional experiments could reinforce our findings. Therefore, we conducted additional experiments including flow cytometry, western blotting, and using kinase-dead mutant mice to further investigate the underlying mechanisms. We carefully addressed every comment by the reviewer as indicated below.

Detailed major concerns:(1) Related to Figure 1.It is imperative to ensure consistency in the number of animals analyzed across the different graphs. The current resolution of the images appears to be low, resulting in unsharp visuals that hinder the interpretation of data beyond the presence of "white dots". To address this issue, it is recommended to enhance the resolution of the images and consider incorporating zoom-in features to facilitate a clearer visualization of the observed differences. Moreover, it would be beneficial to include a complete WB analysis for the cell death pathways analyzed. These adjustments will significantly improve the clarity and interpretability of Figure 1.

Thanks very much for the constructive advice. We carefully checked the number of animals and make sure that the animal number were consistent within different figures. We further updated the figures with incorporating zoom-in features in updated Figure 1, and the resolution of the figures were greatly improved. Western blot analysis were also included in updated Supplementary Figure 1.

(2) Related to Figure 2.It is essential to ensure consistency in the number of animals analyzed across the different graphs, as indicated by n=6 in the figure legend (similar to Figure 1). Additionally, it is crucial to distinguish between male and female subjects in the dot plots to assess any potential gender-based differences, which should be consistent throughout the paper. To achieve this, the dots plot should be harmonized to clearly differentiate between males and females and investigate if there are any disparities between the genders. Moreover, it is imperative to correlate hepatic inflammation with the activation of Kupffer cells, infiltrating monocytes, and/or hepatic stellate cells (HSCs). Therefore, conducting flow cytometry would be instrumental in achieving this correlation. Additionally, the staining for Ki67 appears to be non-specific, showing a granular pattern reminiscent of bile crystals rather than the expected nuclear staining of hepatocytes or immune cells. It is crucial to ensure specific staining for Ki67, and conducting in vitro experiments on primary hepatocytes could further elucidate the proliferation process. These experiments are relatively straightforward to implement and would provide valuable insights into the mechanisms underlying hepatic inflammation and proliferation.

Thanks very much for the helpful advice. First, we corrected the number of animals analyzed in different graphs and make sure that the number of animals listed in the figure legend were consistent with the graphs in all figures. Second, to distinguish the results between male and female mice, blue represents male mice, pink represents female mice, and green represents RIPK1 kinase inactivated mice. The majority of results were obtained from male mice, and our results indicated that there was no difference between male and female mice herein.

The percentages of immune cell subpopulations isolated from mouse liver tissue were determined. The results were consistent with single cell analysis that greater number of macrophages were recruited into the liver tissue in *Ripk1Δhep* upon 12-hour fasting (updated Figure 4F&G).

To confirm the results of Ki67, we first detected the transcriptional expression of Ki67 using real-time qPCR, and the results were consistent with the protein expression measured by immunohistochemical analysis. The percentage of Ki67^+^ cells in liver cells were also detected, and there was significantly more Ki67^+^ cells in *Ripk1Δhep* mouse liver than WT control mouse upon 12-hour fasting. Taken together, our transcriptional analysis, immunohistochemical analysis as well as flow cytometry data indicated that Ki67 expression was higher in *Ripk1Δhep* mice than *Ripk1fl/fl* mice. (updated Figure 2).

(3) Related to Figure 3 & related to Figure 4.The immunofluorescence data presented are not entirely convincing and are insufficient to conclusively demonstrate the recruitment of monocytes. Previous suggestions for flow cytometry studies remain pertinent and are indeed necessary to bolster the robustness of the data and conclusions. Conducting flow cytometry analyses would provide more accurate and quantitative assessments of monocyte recruitment, ensuring the reliability of the findings and strengthening the overall conclusions of the study. Regarding the single-cell RNA sequencing analysis presented in the manuscript, it's worth questioning its relevance and depth of information provided. While it successfully identifies a quantitative difference in the cellular composition of the liver between control and knockout mice, it may fall short in elucidating the intricate interactions between different cell populations, which are crucial for understanding the underlying mechanisms of hepatic inflammation. Therefore, I propose considering alternative bioinformatic analyses, such as CellPhone-CellChat, which could potentially provide a more comprehensive understanding of the cellular dynamics and interactions within the liver microenvironment. By examining the dialogue between different cell clusters, these analyses could offer deeper insights into the functional consequences of Ripk1 deficiency in hepatocytes and its impact on hepatic inflammation during fasting.

Thanks very much for the constructive suggestion. We agree with the reviewer that conducting flow cytometry analyses would provide accurate and quantitative assessments of monocyte recruitment, ensuring the reliability of the findings. Following the advice, both WT and *Ripk1Δhep* mice were fasted for 12 hour and then single hepatic cells were isolated and analyzed by flow cytometry. As indicated in updated Figure 4F&G, the percentage of F4/80^+^CD11b^+^ cells were significantly higher in *Ripk1Δhep* compared with WT control mice, confirming that more monocytes were recruited into the liver.

Additionally, we performed CellChat analysis on the single-cell transcriptomic data. As shown in updated Figures 4H-J, both the number of ligand-receptor pairs and the interaction strength among the eight cell types were significantly increased in *Ripk1Δhep* mice, particularly the interactions between macrophages and other cell types. Network analysis indicated that inflammation and proliferation signals were amplified in *Ripk1Δhep* mice. Consistent with the bulk RNA sequencing data, SAA signaling was upregulated in the hepatocytes of *Ripk1Δhep* mice (updated Figure 4K). SAA has been found to play a role in regulating immune responses and tumor development. Based on these findings, we speculate that fasting-induced liver injury in RIPK1 knockout mice may exacerbate the inflammatory response in liver tissue through enhanced SAA signaling. The above data analysis and interpretation were included in the updated Figure 4&S4 and line 421 - 443.

(4) Related to Figure 5.What additional insights do the data from Figure 5 provide compared to the study published in Nat Comms, which demonstrated that RIPK1 regulates starvation resistance by modulating aspartate catabolism (PMID: 34686667)?

Thank you very much for your constructive suggestion. As noted by the reviewer, this study (PMID: 34686667IF: 14.7 Q1 B1IF: 14.7 Q1) primarily focuses on metabolomic analyses of neonatal mouse brain tissue and MEF cells. The authors propose that Ripk1 regulates starvation resistance by modulating aspartate catabolism.IF: 14.7 Q1 B1

In our study, the global metabolic changes induced by fasting were monitored. Fastinginduced lipolysis in peripheral adipose tissue leads to hepatic lipid accumulation, and excessive deposition of free fatty acids has been shown to induce endoplasmic reticulum (ER) stress in the liver. Data from Figure 5 demonstrate that administering the ER stress inhibitor 4-PBA effectively mitigated fasting-induced liver injury and inflammatory responses in *Ripk1Δhep* mice. Our findings suggest that ER stress plays a critical role in fasting-induced liver injury and inflammation in *Ripk1Δhep* mice.

(5) Related to Figure 6.The data presented in Figure 7 are complementary and do not introduce new mechanistic insights.

Thank you very much for your insightful suggestion. As you mentioned, the AAV-TBG-Cre-mediated liver-specific RIPK1 knockout mice offer complementary validation of the results obtained from *Ripk1Δhep* mice. Moreover, TBG is a promoter that is exclusively expressed in mature hepatocytes, while the ALB promoter is active not only in mature hepatocytes but also in precursor cells and cholangiocytes. Therefore, we think that the inclusion of AAV-TBG-Cre further strengthens our finding that RIPK1 in hepatocytes is responsible for fasting-induced liver injury and inflammatory responses.

(6) Related to Figure 7.The data from Figure 7 suggest that RIPK1 in hepatocytes is responsible for the observed damage. However, it has been previously demonstrated that inhibition of RIPK1 activity in macrophages protects against the development of MASLD (PMID: 33208891). One possible explanation for these findings could be that the overreaction of macrophages to fasting, coupled with the absence of RIPK1 in hepatocytes (an indirect effect), contributes to the observed damage. Considering this, complementing hepatocytes with a kinase-dead version of RIPK1 could be a valuable approach to further refine the molecular aspect of the study. This would allow for a more precise investigation into the specific role of RIPK1's scaffolding or kinase function in response to starvation in hepatocytes. Such experiments could provide additional insights into the mechanisms underlying the observed effects and help delineate the contributions of RIPK1 in different cell types to metabolic stress responses.

Thank you very much for the constructive suggestion. We fully agree with the reviewer that employing a RIPK1 kinase-inactive mutant mice could precisely investigate the specific roles of RIPK1's scaffolding and kinase functions in hepatocyte responses to starvation, respectively. In accordance with this advice, we established a 12-hour fasting model using *Ripk1WT/WT* and *Ripk1K45A/K45A* mice, which were previously established and confirmed with the inactivity of RIPK1 kinase activity. As demonstrated in updated Supplementary Figure 2, these mice did not show significant liver damage or inflammatory responses after 12 hours of fasting. These findings suggest that the liver damage and inflammatory response induced by fasting in *Ripk1Δhep* mice may not be contributed by the kinase activity of RIPK1.

**Reviewer #2 (Public Review):**
Summary:Zhang et al. analyzed the functional role of hepatocyte RIPK1 during metabolic stress, particularly its scaffold function rather than kinase function. They show that Ripk1 knockout sensitizes the liver to cell death and inflammation in response to short-term fasting, a condition that would not induce obvious abnormality in wild-type mice.Strengths:The findings are based on a knockout mouse model and supported by bulk RNA-seq and scRNA-seq. The work consolidates the complex role of RIPK1 in metabolic stress.Weaknesses:However, the findings are not novel enough because the pro-survival role of RIPK1 scaffold is well-established and several similar pieces of research already exist. Moreover, the mechanism is not very clear and needs additional experiments.

We thank the reviewer for the encouraging comments and helpful suggestions. Here we conducted additional experiments including flow cytometry, western blotting, and using kinase-dead mutant mice to further investigate the underlying mechanisms. We carefully addressed every comment by the reviewer as indicated below.

**Recommendations for the authors:**

**Reviewer #1 (Recommendations For The Authors):**
(7) I recommend that the authors consider reassessing their results, particularly with regards to elucidating the dialogue between macrophages and hepatocytes, as this could further strengthen the study's conclusions.

Thank you very much for your constructive suggestion. We conducted additional experiments, including flow cytometry and western blotting, to reassess our findings. Furthermore, to clarify the interactions between cells, we employed CellChat for a more in-depth analysis of the single-cell sequencing results. In the revised manuscript version, changes are highlighted in yellow. In this study, we demonstrated that the specific deletion of RIPK1 in hepatocytes exacerbated the liver's vulnerability to metabolic disturbances, such as short-term fasting and high-fat diet feeding, resulting in increased liver damage, apoptosis, inflammation, and compensatory proliferation. The data indicate that fasting-induced liver injury in RIPK1 knockout mice of hepatic parenchymal cells may exacerbate the inflammatory response in liver tissue through enhanced SAA signaling. In summary, we revealed a novel physiological role of RIPK1 as a scaffold in maintaining liver homeostasis during fasting and other nutritional disturbances.

(8) It would be beneficial for the authors to address the minor weaknesses identified in the study, such as ensuring consistency in the number of animals analyzed across different graphs and enhancing the resolution of images to improve data clarity.

Thank you for the suggestion. In the revised manuscript, we have addressed these minor weaknesses, and we checked the consistency in the number of animals in different graphs, as well as enhanced the resolution of all images.

(9) I encourage the authors to incorporate additional experiments, such as Western blotting and flow cytometry, to provide a more quantitative assessment of the observed effects and enhance the robustness of their conclusions.

Thank you for your insightful suggestion. We completely agree with the reviewer that incorporating flow cytometry and western blotting would strengthen the robustness of our conclusions. We conducted flow cytometry analysis and western blotting and the results were listed in updated Supplementary Figure 1, Figure 2, Figure 4 and Supplementary Figure 4.

(10) Furthermore, the authors may consider conducting complementary experiments, such as rescue experiments involving complementing hepatocytes with a kinase-dead version of RIPK1, to further refine the molecular aspect of the study and elucidate the specific roles of RIPK1's scaffolding or kinase function in response to starvation.

Thank you very much for your constructive suggestion. As shown in updated Supplementary Figure 2, we conducted fasting experiments using RIPK1 kinase-dead mice. These findings suggest that the liver damage and inflammatory response induced by fasting in *Ripk1Δhep* mice may not contributed by the kinase activity of RIPK1.

**Reviewer #2 (Recommendations For The Authors):**
Major:(11) What is the upsteam signal for RIPK1? The study investigated the change induced by short-term fasting which is metabolic stress. Although RIPK1 knockout promotes cell death and inflammation, how it is involved in this condition is unclear. RIPK1 is never reported as a metabolic sensor and its function is typically downstream of TNFR1 as well as other death receptors such as Fas, TRAIL-R1, TRAIL-R2. Thus, it's probable that metabolic stress induces the expression and secretion of some ligand of the above receptors. Although TNFα expression is upregulated on both mRNA and protein levels, it could not be concluded that TNFα is the upsteam signal for RIPK1 because expression difference does not always lead to fuctional role. In addition, a recent study, which is also reference 33, reports that knockout of TNFR1/2 does not protect against 18 h liver ischemia, a condition that is similar to the present study. Therefore, the link between the metabolic fluctuation and RIPK1 function is elusive and should be addressed. The expression difference analysis should be extended to other relevant ligands. A functional study using neutralizing antibodies in RIPK1ΔHep mice is encouraged. At least, this should be discussed in the discussion section.

Thank you very much for your insightful comments. The upstream signals of RIPK1 remains a significant area of scientific inquiry. Fasting, as one of the main causes of metabolic stress, is known to trigger a series of physiological changes, including but not limited to decreased blood glucose levels, hepatic glycogen depletion, increased production of hepatic glucose and ketone bodies, adipose tissue lipolysis, and the influx and accumulation of free fatty lipids in the liver. It is well-established that the elevated lipid influx and hepatic accumulation during fasting may cause lipotoxicity stress for liver. To investigate whether the elevated free fatty acids influx might act as the signal to induce cytotoxicity, we isolated primary hepatocytes but observed that a significant number of cells underwent spontaneous death during the isolation and perfusion processes. To address this question, we utilized CRISPR-Cas9 technology to generate *Ripk1-/-* AML12 cells, as illustrated in Author response image 1A.

To mimic hepatic lipid accumulation induced by short-term fasting, we treated the cells with palmitic acid (PA) or oleic acid (OA) for 12 hours in vitro. Our results indicated a significant increase in cell death among *Ripk1-/-* AML12 cells after PA treatment compared to WT control cells (Author response image 1B). As shown in Author response image 1C, we also observed a marked increase in caspase-3 activity in *Ripk1-/-* AML12 cells following PA treatment.

Collectively, our results highlight the crucial role of RIPK1 in hepatocytes in maintaining the liver's adaptive capacity to counteract lipotoxicity induced by metabolic stress. These in vitro results were not included in the manuscript; however, we addressed them in the discussion section (line 593 - 597). If the reviewer suggest, we would like to incorporate in our manuscript.

(12) What is the exact relationship between ER stress and RIPK1? In Figure 5A and Figure 6B, Ripk1 knockout only slightly promotes the expression of ER stress markers. The evidence of RIPK1 leading to ER stress is limited in the literature and poorly supported in this study. Also in reference 33, the hypothesis is proposed that ER stress leads to death receptor upregulation and activation, which induces RIPK1 activation. Although the ER stress inhibitor showed good efficacy in rescue experiments, it could not determine whether RIPK1 deficiency leads to ER stress-associated phenotype or ER stress leads to death receptor activation and RIPK1 deficiency-associated phenotype. If RIPK1 deficiency leads to ER stress, the possible mechanism should be investigated.

Thank you very much for your insightful comments. As the reviewer noted, the specific relationship between endoplasmic reticulum (ER) stress and RIPK1 remains unclear. However, our data, along with findings from other studies (Piccolis M et al., Mol Cell. 2019; Geng Y et al., Hepatol Int. 2021), suggest that fasting-induced lipolysis in peripheral adipose tissue leads to hepatic lipid accumulation. Additionally, excessive deposition of free fatty acids has been shown to induce ER stress in the liver. One possible explanation is that ER stress may trigger the upregulation and activation of death receptors, and the scaffold function of RIPK1 may play a protective and checkpoint role in this process. ER stress during the fasting might locate upstream of RIPK1. This could help explain why short-term fasting results in liver damage in *Ripk1Δhep* mice while control mice remain unaffected. Moreover, the inhibition of ER stress using 4-PBA can effectively alleviate this damage.

Minor:(13) The study starts directly from functional experiments. However, it should be firstly explored whether RIPK1 expression or activation is modulated in wild-type mice.

Thank you very much for your insightful observation. Previous studies showed that RIPK1 deficiency in hepatocytes does not impact the growth and development of mice, indicating that RIPK1 is dispensable for proper liver development and homeostasis (Filliol A et al., Cell Death Dis. 2016). Furthermore, we did not observe any changes in RIPK1 levels in wild-type mice induced by fasting across different experimental batches. In our bulk transcriptomic analysis, the expression of RIPK1 was not changed before and after 12-hour fasting in *Ripk1fl/fl* mice. Therefore, we focused our attention on the function of RIPK1 and started our study directly with functional experiments.

(14) Knockout of RIPK1 deprived both its scaffold function and kinase function. It is encouraged to explore whether blocking RIPK1 kinase activity influences the outcome of metabolic stress.

Thank you for your insightful suggestion. To investigate the role of RIPK1 kinase activity in response to metabolic stress, we added fasting experiments using RIPK1 kinaseinactive mice in the updated Supplementary Figure 2, in which blocking RIPK1 kinase activity does not affect the outcome of metabolic stress.

(15) In Figure 1, the number of TUNEL+ cells is about 2 times of c-casp3. What is the possible reason?

Thank you for your careful reading. Indeed, the number of TUNEL^+^ cells in Figure 1 is twice that of cleaved-caspase-3^+^ cells. There are two possible reasons. First, we speculate that this discrepancy may be attributed to the higher sensitivity of the TUNEL assay compared to the cleaved-caspase-3 assay. Secondly, TUNEL assay detects DNA fragmentation, indicating that these cells are in a pre-apoptotic state or poised to undergo apoptosis. In contrast, cleaved-caspase-3 specifically identifies cells that have already committed to the apoptotic pathway, whereas TUNEL assay could detects all types of apoptosis, but the mechanisms of apoptosis may involve more than just cleaved-caspase3.

(16) Infiltrated innate immune cells could lead to hepatocyte death. Is the hepatocyte death in this study partially caused by immune cells?

Many thanks for the advice. As outlined in the response to the 11th comment from the second reviewer, our findings indicate that metabolic stress induced by short-term fasting is the primary cause of hepatocyte death. Additionally, we demonstrate that infiltrated innate immune cells may also play a partial role in hepatocyte death through subsequent cascade reactions.

(17) Could the in vivo results be consolidated by in vitro experiments on primary mouse hepatocytes? This would be helpful to answer question 4.

Thank you for your helpful comments. As demonstrated in the response to the 11th comment by the second reviewer, we attempted to conduct in vitro experiments using primary hepatocytes. However, during the isolation and perfusion processes, we observed that a significant number of cells underwent spontaneous death. To address this issue, we utilized CRISPR-Cas9 technology to generate *Ripk1-/-* AML12 cells, in which a significant increase in cell death among *Ripk1-/-* AML12 cells after palmitic acid (PA) treatment compared to WT control cells. We also observed a marked increase in caspase-3 activity in *Ripk1-/-* AML12 cells following PA treatment.

(18) RIPK1 scaffold function is associated with NF-kB signal. Is NF-kB signal transduction influenced by Ripk1 deficiency? If so, to what extent does it contribute to the observed phynotype? If not, what is the direct downstream effect of Ripk1 deficiency?

Thank you very much for your insightful perspective. As reported by Clucas J et al., RIPK1 serves as a scaffold for downstream NF-κB signaling through the ubiquitin chains generated by its ubiquitination (Clucas J et al., Nat Rev Mol Cell Biol. 2023). The deficiency of RIPK1 in hepatic parenchymal cells can disrupt NF-κB signaling and impair its pro-survival functions, resulting in increased cell death in response to stress. Our current findings suggest that the RIPK1-NF-κB axis serves as a crucial scaffold platform essential for the liver's adaptation to metabolic fluctuations. Any inappropriate inactivation or deletion of components within this scaffold disrupts the delicate balance between cell death, inflammation, and normal function, making the liver susceptible to metabolic changes, ultimately leading to liver damage, hepatic inflammation, and compensatory proliferation.

(19) In Figure 6B, the 'RIP' should be changed to 'RIPK1'.

Thank you for your careful observation. We have corrected "RIP" to "RIPK1" in updated Figure 6B.

(20) For Western blot results, the blot height should be at least the lane width to reveal additional signals and the molecular weight as well as unspecific signals should be denoted.

Thank you for your valuable advice. We appreciate your suggestions regarding the western blot results. We went through the previous western blot results and did not find any additional nonspecific signals. We added the molecular weights in the updated figures Figure 5, Figure 6 and Supplementary Figure 1.